# MARTX toxin of *Vibrio vulnificus* induces RBC phosphatidylserine exposure that can contribute to thrombosis

Han Young Chung[1,2], Yiying Bian[3], Kyung-Min Lim[4], Byoung Sik Kim [ID][5] & Sang Ho Choi [ID][1,2] ✉

*V. vulnificus*-infected patients suffer from hemolytic anemia and circulatory lesions, often accompanied by venous thrombosis. However, the pathophysiological mechanism of venous thrombosis associated with *V. vulnificus* infection remains largely unknown. Herein, *V. vulnificus* infection at the sub-hemolytic level induced shape change of human red blood cells (RBCs) accompanied by phosphatidylserine exposure, and microvesicle generation, leading to the procoagulant activation of RBCs and ultimately, acquisition of prothrombotic activity. Of note, *V. vulnificus* exposed to RBCs substantially upregulated the *rtxA* gene encoding multifunctional autoprocessing repeats-in-toxin (MARTX) toxin. Mutant studies showed that *V. vulnificus*-induced RBC procoagulant activity was due to the pore forming region of the MARTX toxin causing intracellular $Ca^{2+}$ influx in RBCs. In a rat venous thrombosis model triggered by tissue factor and stasis, the *V. vulnificus* wild type increased thrombosis while the *ΔrtxA* mutant failed to increase thrombosis, confirming that *V. vulnificus* induces thrombosis through the procoagulant activation of RBCs via the mediation of the MARTX toxin.

Infection of *V. vulnificus*, an opportunistic human pathogen and a causative agent of foodborne disease, is characterized by rapid disease progression leading to hemolysis and fatal septicemia with high mortality rates within a few days[1,2]. The pathogenesis of *V. vulnificus* infection is also often accompanied by circulatory disorders such as venous thrombosis, which brings about tissue destruction and limb amputation[3–5]. These pathological features imply that *V. vulnificus* infection may induce intravascular coagulation[6], which causes blood clotting and blockade of small blood vessels. Also, the pathological features suggest that the prevention of venous thrombosis is an attractive strategy to alleviate the symptoms of *V. vulnificus* infection. However, little is known about the virulence factor(s) of *V. vulnificus* responsible for the development of venous thrombosis and its molecular mechanisms.

Normally, thrombosis, such as arterial thrombosis, occurs via platelet activation under high blood flow[7]. However, recent studies demonstrated that thrombosis, especially in the vein, also occurs through the increased procoagulant activity of human RBCs under low blood flow[8,9]. Procoagulant activity is often increased along with the elevation of the intracellular calcium $[Ca^{2+}]_i$ level, and caspase-3 and scramblase activities by various external stimuli such as mechanical and chemical stress, resulting in the externalization of an anionic phospholipid, phosphatidylserine (PS) of RBCs[9,10]. Along with PS exposure, RBCs often shed the PS-bearing microvesicles (MVs; less than 1 μm in diameter) by exocytosis[11] and undergo progressive morphological changes from normal discocytes to echinocytes[12]. The PS exposure of RBCs provides sites for the assembly of prothrombinase and tenase complexes, which facilitates thrombin generation and

[1]National Research Laboratory of Molecular Microbiology and Toxicology, Department of Agricultural Biotechnology, Seoul National University, Seoul 08826, Republic of Korea. [2]Center for Food and Bioconvergence, Seoul National University, Seoul 08826, Republic of Korea. [3]School of Public Health, China Medical University, Shenyang 110122, People's Republic of China. [4]College of Pharmacy, Ewha Womans University, Seoul 03760, Republic of Korea. [5]Department of Food Science and Engineering, Ewha Womans University, Seoul 03760, Republic of Korea. ✉e-mail: choish@snu.ac.kr

enhances prothrombotic activity to prompt blood clotting and, ultimately, venous thrombosis[13].

Among multiple virulence factors of *V. vulnificus*, the multifunctional autoprocessing repeats-in-toxin (MARTX) toxin, which is encoded by the *rtxA* gene, is the most crucial exotoxin responsible for the pathogenicity of *V. vulnificus*[14,15]. As a very large protein toxin, the MARTX toxin forms a pore on the host cell membrane using its repeats-containing regions and then delivers its own multiple cytotoxic or cytopathic effector domains into the cytosol of epithelial cells[16]. A mutant *V. vulnificus* defective in the MARTX toxin production was less cytotoxic and lethal when compared with the isogenic wild type (WT)[14,15]. However, it remains unelucidated whether the MARTX toxin also plays a role in venous thrombosis caused by *V. vulnificus* infection.

Here, we show that *V. vulnificus* induces shape changes, PS exposure, and MV generation in RBCs, along with the elevation of the $[Ca^{2+}]_i$ level, ultimately leading to the procoagulant and prothrombotic activity of RBCs. Especially, *V. vulnificus* infecting RBCs substantially upregulated *rtxA*, illuminating that the pore-forming activity of the MARTX toxin induced the cytopathic changes of RBCs. Finally, we conducted rat model studies in vivo to confirm the role of the MARTX toxin in *V. vulnificus*-associated thrombosis.

## Results

### Morphological changes of RBCs infected with *V. vulnificus* at the sub-hemolytic level

To examine the possible changes of human red blood cells (RBCs) before hemolysis, an optimal titer and incubation time for sub-hemolytic infection of *V. vulnificus* were screened. The RBCs freshly isolated from healthy volunteers were infected with *V. vulnificus* MO6-24/O (Supplementary Table 1), and then the multiplicity of infection (MOI)-dependent and incubation time-dependent hemolysis was observed (Fig. 1a, b). When incubated for 1 h, *V. vulnificus* induced only minimal hemolysis (<20%) at an MOI lower than 10, but severe hemolysis (up to 80%) was observed at an MOI of 25 (Fig. 1a). Similarly, *V. vulnificus* lysed less than 20% of RBCs at an MOI of 5 or 10 and incubation for up to 1 h (Fig. 1b). Therefore, infections of RBCs with *V. vulnificus* at an MOI up to 10 and incubation for shorter than 1 h

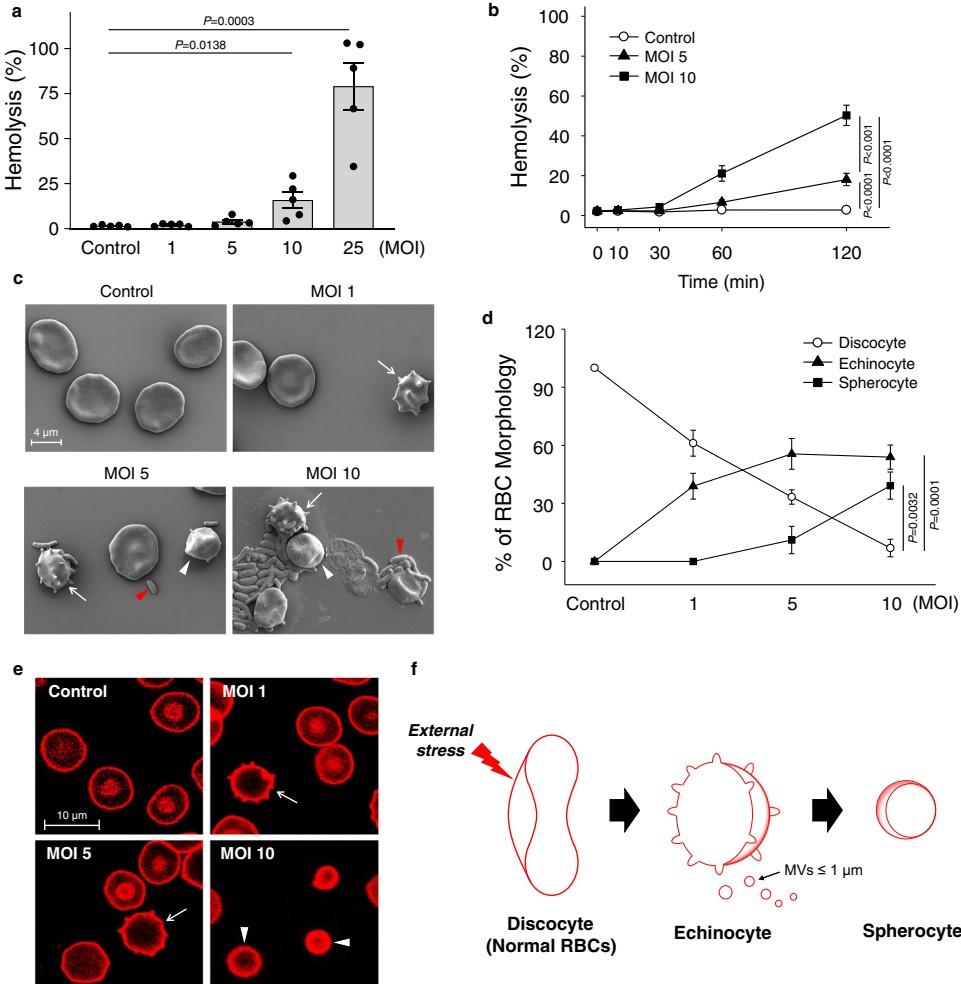

**Fig. 1 | Infection of human RBCs with *V. vulnificus* induces hemolysis and shape change. a**, **b** RBCs were infected with *V. vulnificus* at various MOIs and incubated for 1 h (**a**) (*n* = 5, two-tailed Student's *t*-test) or for different times (**b**) (*n* = 6, two-way analysis of variance (ANOVA) followed by Duncan's multiple range test) as indicated. **c** RBCs were infected with *V. vulnificus* at various MOIs and incubated for 1 h and shape changes were analyzed by SEM. The red arrowhead indicates *V. vulnificus*; white arrow, echinocyte; and the white arrowhead, spherocyte. The scale bar is presented. **d** Different shapes of RBCs were quantified from the SEM images obtained from six independent experiments and were expressed as the percent of the specific shapes per total RBCs (*n* = 6, two-tailed Student's *t*-test). **e** RBCs were infected with *V. vulnificus* at various MOIs and incubated for 1 h. The representative images by confocal microscopy among five independent experiments were presented. Different shapes of RBCs are marked with the same symbols used for Fig. 1c. Scale bar is presented. **f** A schematic diagram represents a typical morphological change of RBCs under external stress. The means ± SE were calculated from at least five independent experiments. Control, uninfected; MOI multiplicity of infection, SEM scanning electron microscopy, and MV microvesicle.

were defined as the sub-hemolytic level infections in the following experiments.

Interestingly, *V. vulnificus* (red arrowheads) infection at the sub-hemolytic level induced the morphological changes of the RBCs (discocyte) as analyzed with scanning electron microscopy (SEM) (Fig. 1c) and transmission electron microscopy (TEM) (Supplementary Fig. 1). As the MOI increased up to 10, echinocytes (white arrows) characterized by a shell-like appearance with evenly spaced thorny projections, followed by spherocytes (white arrowheads) with a small-sized round shape, were observed (Fig. 1c, d). The morphological changes of discocyte infected with *V. vulnificus* into echinocyte and then spherocyte were further confirmed with confocal microscopy (Fig. 1e). The morphological changes of RBCs infected with *V. vulnificus* into the echinocytes accompanied with numerous MVs generate spherocytes (Fig. 1f), as often observed under various mechanical and chemical stress[12].

## MV generation and phosphatidylserine (PS) exposure of RBCs

MV generation of the RBCs infected with *V. vulnificus* at the sub-hemolytic level were further investigated. Flow cytometry analysis demonstrated that the RBCs uninfected and infected with *V. vulnificus* at an MOI of 5 are observed as a uniform size (mostly RBCs) and two clearly different sizes (RBCs and MVs), respectively (Fig. 2a, top left). Along with MV generation, PS exposures increased from 2.1% (control) to 49.2% of RBCs upon infection with *V. vulnificus* at an MOI of 5 (Fig. 2a, top right). In fact, PS exposures of the RBCs increased with *V. vulnificus* infection in an MOI-dependent manner (Fig. 2a, middle right). Similarly, MV generation and their PS exposure also increased in an MOI-dependent manner (Fig. 2a, bottom). Confocal microscopy analysis demonstrated that the uninfected RBCs were not stained with Annexin V specific for the exposed PS (Fig. 2b). However, the outer leaflet of the echinocyte and spherocyte membranes were positively stained (Fig. 2b), confirming PS exposure of RBCs upon *V. vulnificus* infection. Notably, infection of RBCs with *V. vulnificus* increased MV generation and PS exposure not only in an MOI-dependent but also in an incubation time-dependent manner (Supplementary Fig. 2a, b). The results confirmed that the morphological changes of RBCs infected with *V. vulnificus* into spherocytes at the sub-hemolytic level are accompanied by MV generation and PS exposure.

## Procoagulant activity of RBCs

PS exposure of RBCs under mechanical and chemical stress is often accompanied by procoagulant activity[9]. Interestingly, RBCs infected with *V. vulnificus* also revealed procoagulant activity along with the elevation of the intracellular $Ca^{2+}$ $[Ca^{2+}]_i$ level (Fig. 2c), $Ca^{2+}$-dependent caspase-3 activity (Fig. 2d), and then scramblase activity (Fig. 2e) in an MOI-dependent manner. This result indicated that the increased PS exposure of the RBCs, observed upon infection with *V. vulnificus* (Fig. 2a), is also accompanied by increased procoagulant activity. To confirm this relationship between PS exposure and procoagulant activity, RBCs were pretreated with specific inhibitors of caspase-3, Z-DEVD-FMK (VI), and Q-VD-OPh (Q) before being infected with *V. vulnificus*. When infected with *V. vulnificus*, the pretreated RBCs revealed significantly reduced PS exposure and procoagulant activity (Fig. 2f), confirming that PS exposure is caused by the procoagulant activity of RBCs.

## Prothrombotic activity of RBCs

The increased procoagulant activity could result in functional alterations of RBCs, such as promoting their prothrombotic activity, including thrombin generation, adherence to endothelial cells, and self-aggregation[9,13,17,18]. When infected with *V. vulnificus*, RBCs increased thrombin generation in an MOI-dependent and an incubation time-dependent manner (Fig. 2g). When the exposed PS in RBCs was blocked with purified Annexin V, the procoagulant activity of

*V. vulnificus* was attenuated significantly, suggesting that PS exposure plays a key role in *V. vulnificus*-induced procoagulant activity (Supplementary Fig. 3). RBCs were pre-infected with *V. vulnificus* at different MOIs, and then their adherence and aggregation to human umbilical vein endothelial cells (ECs) were analyzed by fluorescence microscopy. Notably, the RBCs pre-infected with more *V. vulnificus* (MOI of 5 to 10) revealed more adherence (white arrows) and aggregation (yellow arrowheads) to ECs (Fig. 2h). Indeed, *V. vulnificus* infection led RBCs to aggregate themselves in an MOI-dependent manner even in the absence of ECs (Fig. 2i). The combined results suggest that *V. vulnificus* infection at the sub-hemolytic level results in the morphological changes of RBCs, with increased procoagulant activity leading to increased prothrombotic activity.

## MARTX toxin responsible for the functional changes of RBCs

Since *V. vulnificus* infection increased procoagulant activity (Fig. 2a–f and Supplementary Fig. 2a, b), it is reasonable to assume that the *V. vulnificus* virulence factor(s) responsible for procoagulant activity could be specifically induced upon exposure to RBCs. Indeed, the sequencing analyses of the whole transcriptome of *V. vulnificus* unexposed or exposed to RBCs identified a total of 121 differentially expressed genes (DEGs; with |$\log_2$ fold change| ≥1.0, and *P* value ≤0.05) (Fig. 3a and Supplementary Table 2). Most of the 36 genes upregulated are involved in the transport and metabolisms of various nutrients, including iron. Interestingly, the only upregulated exotoxin gene is *rtxA* (VVMO6_03947, $\log_2$ fold change = 1.20, *P* value = 2.72E-43) encoding the MARTX toxin. To examine the effects of the MARTX toxin on procoagulant activity, effects of the *rtxA* deletion in *V. vulnificus* on the changes of PS exposure and MV generation of the RBCs were primarily characterized. The *ΔrtxA* mutant showed significantly reduced PS exposure and MV generation (Fig. 3b, c). Furthermore, the *ΔrtxA* mutant induced significantly less elevation of the $[Ca^{2+}]_i$ level, scramblase activity, and thrombin generation in the RBCs than the parental WT (Fig. 3d–f). Notably, another *rtxA* mutant strain (*rtxA::nptl*), of which the *rtxA* gene was inactivated by insertional rather than deletional mutation, revealed similarly attenuated procoagulant activity of RBCs (Supplementary Fig. 4a–c). Importantly, the attenuation was recovered in a revertant strain, indicating that the effects of the *rtxA* mutation on the cytopathic changes of RBCs were not due to the polar effects (Supplementary Fig. 4b, c). These results confirmed that the MARTX toxin plays a crucial role in the increase of the procoagulant activity of RBCs infected with *V. vulnificus*.

## Pore-forming regions of MARTX toxin responsible for the functional changes of RBCs

To further identify functional domains of the multifunctional MARTX toxin responsible for the increased procoagulant activity of RBCs, an effector-free (EF) *V. vulnificus* mutant (*EF-rtxA*) was constructed (Fig. 3g). Of note, the *EF-rtxA* strain produces an EF-MARTX toxin that does not carry the effector domains but is still able to generate a pore on the intoxicated host cell membrane. The pore-forming activity of the *EF-rtxA* strain was proved by the bleb formation in human epithelial cells (Supplementary Fig. 4d), as reported previously[19]. When infected with either the *V. vulnificus* WT or *EF-rtxA* strains, RBCs revealed no statistical differences in the changes of PS exposure, $[Ca^{2+}]_i$ level, and thrombin generation (Fig. 3h–j). Altogether, the results indicate that the pore-forming regions, not the effector domains, of the MARTX toxin of *V. vulnificus* are responsible for the cytopathic changes of RBCs with their enhanced procoagulant activity.

## Venous thrombosis of rat

Prior to confirming the effects of the MARTX toxin on the RBCs in vivo, the changes in PS exposure and thrombin generation in rat RBCs in

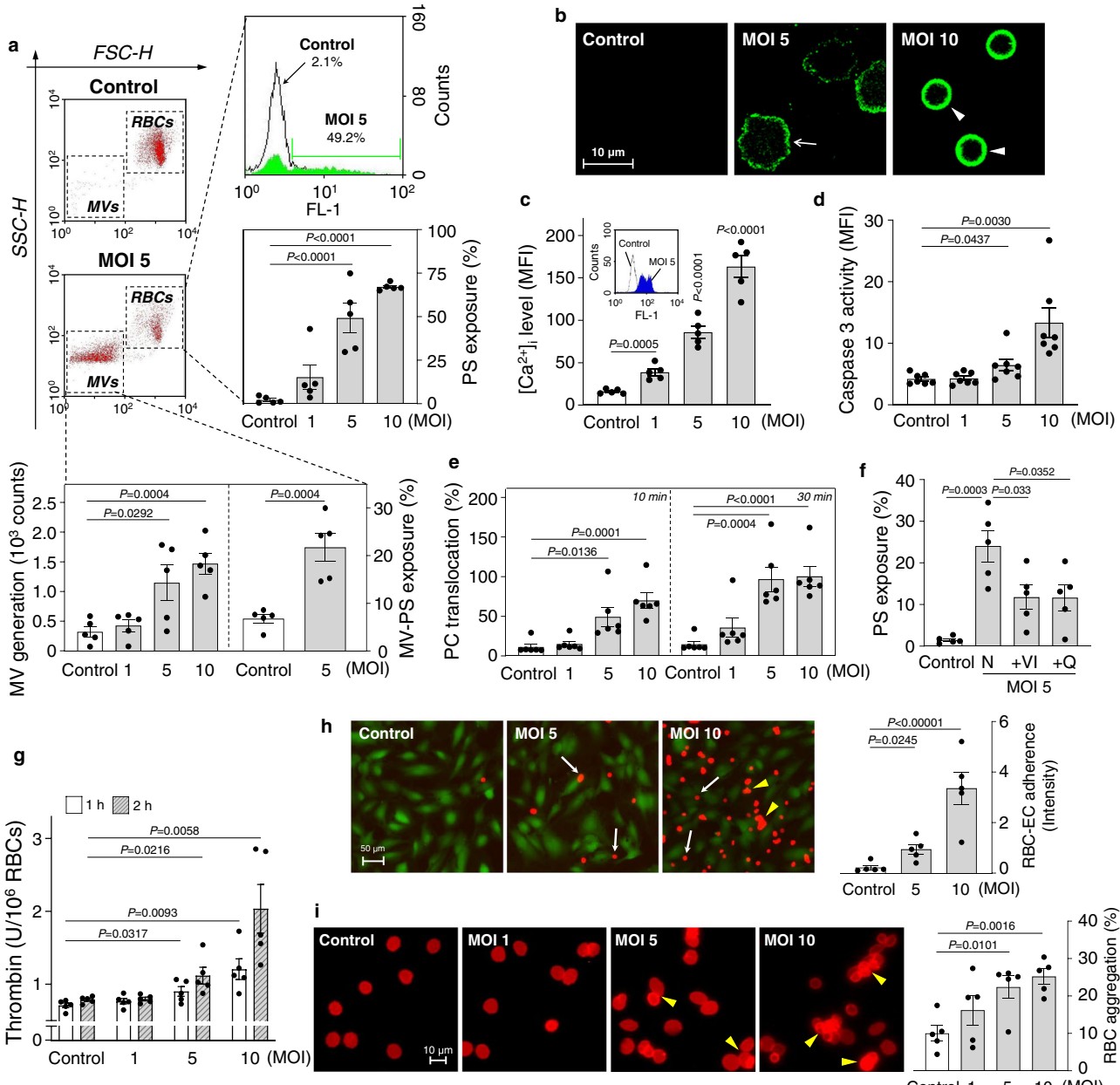

**Fig. 2 | Infection of human RBCs with *V. vulnificus* increases MV generation, PS exposure, and procoagulant and prothrombotic activity. a**, **b** RBCs were infected with *V. vulnificus* at various MOIs and incubated for 1 h. **a** RBCs and MV were separately analyzed based on forward-scatter height (FSC-H) and side-scatter height (SSC-H) by flow cytometry (top left). The extents of PS exposures of the RBCs are determined and presented as a histogram with fluorescence FL-1 (top right) and as bar graphs (middle right) (*n* = 5, two-tailed Student's *t*-test). Consistently, MV generated from RBC membranes (bottom left) and PS exposures of the MVs (bottom right) (*n* = 5, two-tailed Student's *t*-test) are presented. **b** PS exposures of the RBCs stained with Annexin V-FITC were analyzed by confocal microscopy and different shapes of RBCs are marked with the same symbols used for Fig. 1c. The representative images among five independent experiments were presented. The scale bar is presented. **c** Intracellular $Ca^{2+}$ $[Ca^{2+}]_i$ level in the RBCs infected with *V. vulnificus* for 1 h are presented as bar graphs and as a histogram with fluorescence FL-1 (**c**, insert) (*n* = 5, two-tailed Student's *t*-test). **d**–**g** RBCs were

infected with *V. vulnificus* and then caspase-3 activity (**d**) (*n* = 7, two-tailed Student's *t*-test), PC translocation to assess scramblase activity (**e**) (*n* = 6, two-tailed Student's *t*-test), PS exposures in the presence of various caspase inhibitors (**f**) (*n* = 5, two-tailed Student's *t*-test), and thrombin generation (**g**) (*n* = 5, two-tailed Student's *t*-test) were determined. **h** RBCs pre-infected with *V. vulnificus* for 30 min were moved to ECs (green fluorescence), and then RBCs (red fluorescence) adhered to ECs (white arrows) and aggregation (yellow arrowheads) were analyzed by fluorescence microscopy. The relative intensity of the red fluorescence adhered to ECs was presented as a bar graph (*n* = 5, two-tailed Student's *t*-test). **i** RBCs were infected with *V. vulnificus* for 1 h and self-aggregation of the RBCs (yellow arrowheads) were analyzed by fluorescence microscopy. Percentages of the RBC aggregation were presented as bar graphs (*n* = 5, two-tailed Student's *t*-test). The means ± SE were calculated from at least five independent experiments. Control, uninfected; PC phosphatidylcholine, N no inhibitor, VI Z-DEVD-FMK, and Q Q-VD-OPh.

response to *V. vulnificus* infection in vitro were examined as bridge studies. The rat RBCs infected with the sub-hemolytic level of *V. vulnificus* (MOI 1, 5, or 10) in vitro increased PS exposure and thrombin generation in an MOI-dependent manner (Fig. 4a, b), which are similar

to the changes observed in human RBCs (Fig. 2a, g). Next, rats were intravenously (IV) infected with either the *V. vulnificus* WT or *ΔrtxA* for 1 h, and then thrombin generation from the whole blood cells ex vivo were examined. The thrombin generation level of the rats infected with

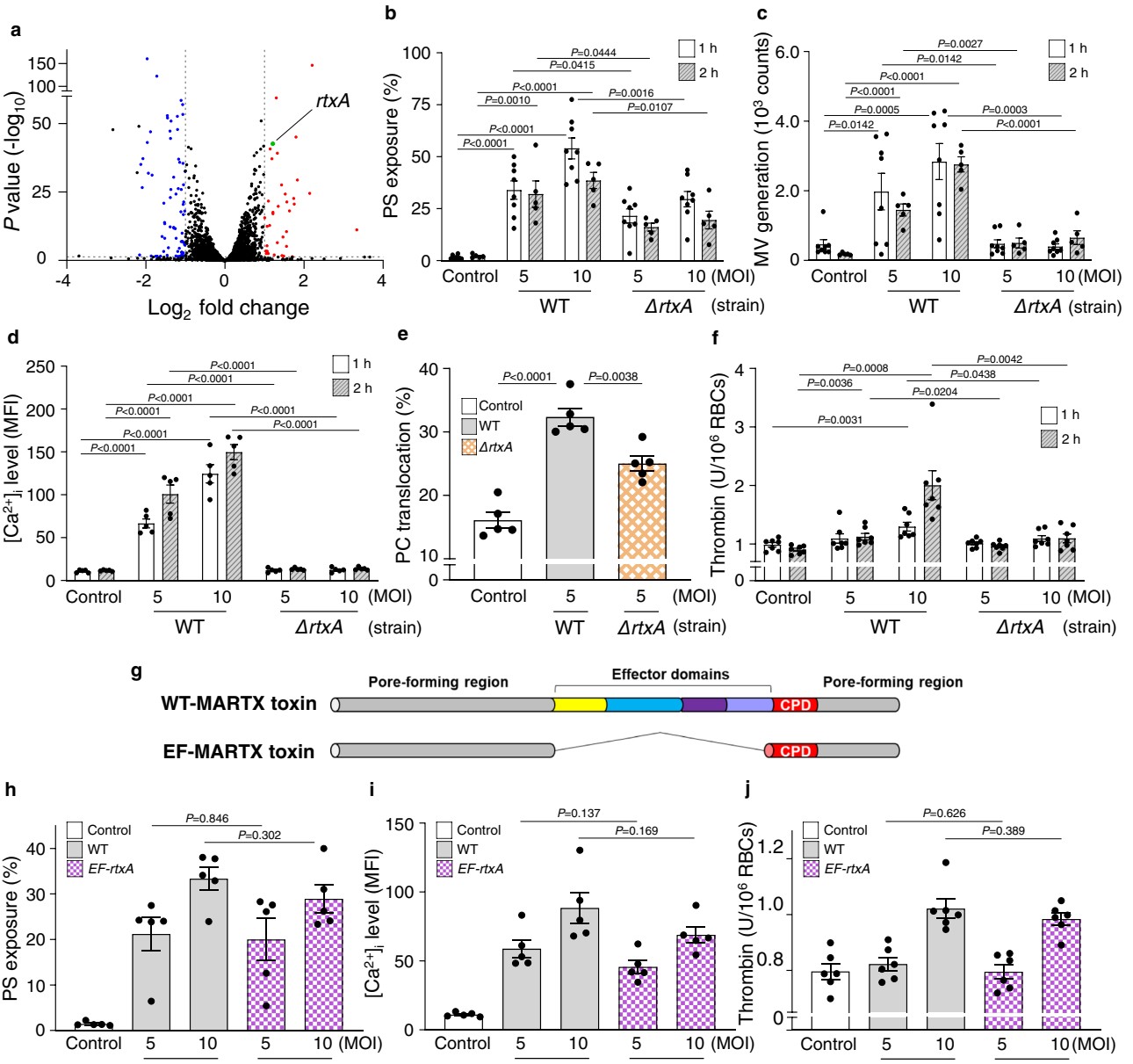

**Fig. 3 | The *V. vulnificus* MARTX toxin is responsible for the procoagulant and prothrombotic activity of human RBCs. a** A volcano plot representing differentially expressed genes in *V. vulnificus* exposed to RBCs. The genes significantly upregulated or downregulated (|log₂ fold change| ≥1.0, and *P* value ≤0.05) are shown in red or blue dots, respectively. A *rtxA* gene is indicated by a green dot. **b**–**f** RBCs were infected with either *V. vulnificus* WT or *ΔrtxA* at various MOIs for 1 or 2 h as indicated. Then, PS exposure and MV generation (**b**, **c**, respectively) (*n* = 8 in 1 h, *n* = 5 in 2 h, two-tailed Student's *t*-test), intracellular Ca²⁺ [Ca²⁺]ᵢ level (**d**) (*n* = 5, two-tailed Student's *t*-test), PC translocation to assess scramblase activity (**e**) (*n* = 5, two-tailed Student's *t*-test), and thrombin generation (**f**) (*n* = 7, two-tailed Student's *t*-test) of the RBCs were determined. **g** Schematic diagrams of

the WT-MARTX toxin and the EF-MARTX toxin. Different colors represent the pore-forming region and each effector domain. **h**–**j** RBCs were infected with either *V. vulnificus* WT or *EF-rtxA* strains at the indicated MOIs for 1 h. Then, PS exposure (**h**) (*n* = 5, two-tailed Student's *t*-test), [Ca²⁺]ᵢ level (**i**) (*n* = 5, two-tailed Student's *t*-test), and thrombin generation (**j**) (*n* = 6, two-tailed Student's *t*-test) were determined. The means ± SE were calculated from at least five independent experiments. WT wild type, EF effector-free, *ΔrtxA* a mutant producing no MARTX toxin, *EF-rtxA* a mutant producing the EF-MARTX toxin, control uninfected, MOI multiplicity of infection, PS phosphatidylserine, MV microvesicle, PC phosphatidylcholine, and CPD cysteine protease domain.

the *ΔrtxA* mutant was lower than that infected with the WT, even close to that uninfected (Fig. 4c), indicating that the MARTX toxin is the virulence factor responsible for thrombin generation from the whole blood cells of the rats. In order to observe the morphological change of RBCs following *V. vulnificus* infection in vivo, blood was collected 1 h after IV injection of WT and *ΔrtxA* mutant to rats. Representative image of RBCs was observed under scanning electron microscopy (Fig. 4d). Consistently with in vitro results (Fig. 1c, e), administration of WT induced morphological changes of RBCs into echinocytes

(white arrows), implying that PS-bearing microvesicles were formed from RBCs in vivo following *V. vulnificus* infection. On the other hand, injection of the *ΔrtxA* mutant did not change the shape compared to the untreated control. Finally, effects of *V. vulnificus* infection on thrombus formation in vivo were examined using a rat venous thrombosis model (Fig. 4e). When rats were IV infected with the *V. vulnificus* WT, and then injected with the thromboplastin, thrombus formation increased in an incubation time-dependent and an infectious dose-dependent manner (Fig. 4f, g). In contrast, rats infected

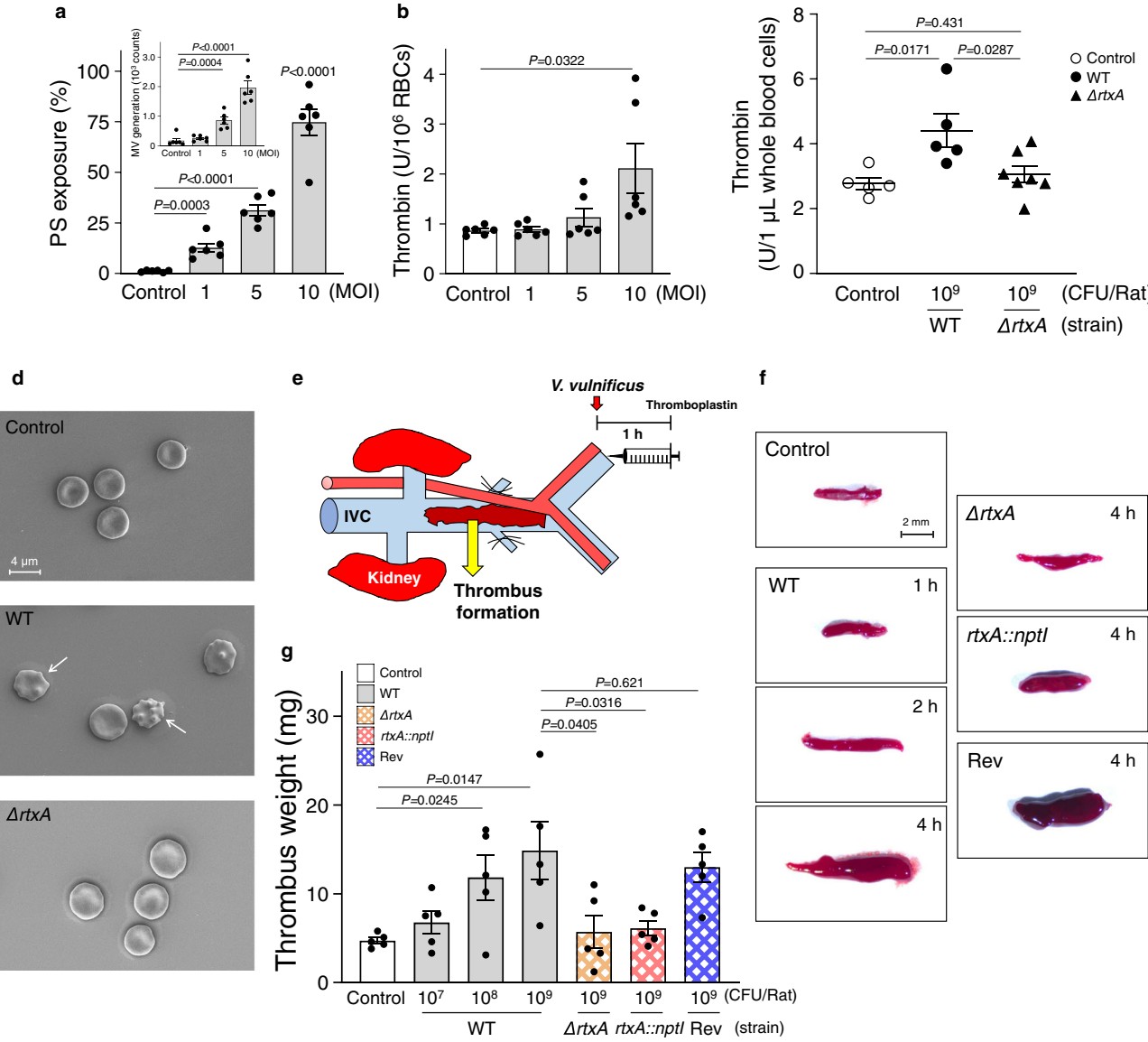

**Fig. 4 | The V. vulnificus MARTX toxin is responsible for rat venous thrombosis.** **a**, **b** Rat RBCs were infected with *V. vulnificus* at the indicated MOIs and incubated for 10 min, and then PS exposure and MV generation (**a**, insert) (**a**) (*n* = 6, two-tailed Student's *t*-test), and thrombin generation (**b**) (*n* = 6, two-tailed Student's *t*-test) were determined. **c** Rats were infected IV with either *V. vulnificus* WT or *ΔrtxA* mutant (10⁹ CFU/Rat) for 1 h, and then whole blood cells were collected and used to determine thrombin generation ex vivo (*n* = 5 in control, *n* = 5 in WT, *n* = 7 in *ΔrtxA* mutant, two-tailed Student's *t*-test). **d** Rats were infected IV with either *V. vulnificus* WT or *ΔrtxA* mutant (10⁸ CFU/Rat) for 1 h, and then whole blood cells were collected and removed platelet-rich plasma and buffy coat to observe RBCs morphology using SEM. The representative images by SEM images among five independent experiments were presented. The white arrow indicates echinocyte. The scale bar is presented. **e** A schematic diagram presents the isolation of thrombus formed in 16 mm of inferior vena cava (IVC) exposed after 1 h IV infection with *V. vulnificus* and subsequent injection with thromboplastin. **f** Rats were infected IV with *V. vulnificus* WT, *ΔrtxA*, *rtxA::nptI* mutant, or revertant strain (10⁷ CFU/Rat) as described above and then incubated for different times as indicated, and then thrombus formed in the IVC were analyzed by stereomicroscopy. The scale bar is presented. **g** Rats were infected IV with *V. vulnificus* WT, *ΔrtxA*, *rtxA::nptI* mutant, or revertant strain for 1 h at various doses as indicated, and then weights of the formed thrombus in the IVC were determined (*n* = 5, two-tailed Student's *t*-test). The means ± SE were calculated from at least five independent experiments. WT wild type, *ΔrtxA* a mutant producing no MARTX toxin, *rtxA::nptI* a mutant producing no MARTX toxin, *nptI* aminoglycoside 3'-phosphotransferase gene, Rev a revertant producing the WT-MARTX toxin, SEM scanning electron microscopy, control, uninfected, MOI multiplicity of infection, and IVC inferior vena cava.

with the deletion mutant (*ΔrtxA*) as well as the insertion mutant (*rtxA::nptI*) did not show any significant increase in thrombus formation when compared with the uninfected control. Furthermore, the revertant of the insertion mutant (*rtxA::nptI*) rescued the ability of *V. vulnificus* WT to increase thrombus formation (Fig. 4f, g), as consistent with in vitro data (Supplementary Fig. 4b, c). These results support our conclusion that the MARTX toxin of *V. vulnificus* is mainly responsible for the procoagulant activity of RBCs and thrombus formation in vivo.

## Discussion

Various pathogens can cause venous thrombosis[3,4,6,20,21] and severe hemolytic anemia[22,23] as reported in many clinical case reports. However, studies on the morphological change of human RBCs caused by these pathogens or its implication in venous thrombosis have not been reported. Here, we demonstrated that *V. vulnificus* infection at the sub-hemolytic level induces remarkable morphological changes in RBCs from normal discocytes to echinocytes and further to the spherocytes (Fig. 1c–f). Along with the morphological changes of RBCs, the

procoagulant activity of RBCs enhances PS exposure and MV generation, ultimately promoting the risk of venous thrombosis. This was fully demonstrated by our in vivo study, where normal healthy rats infected with *V. vulnificus* showed increased thrombin generation of RBCs and thrombus formation while those infected with the *rtxA* mutants did not show any increase (Fig. 4c, f, g). Moreover, the revertant of the *rtxA* insertion mutant (*rtxA::nptI*) rescued the ability of *V. vulnificus* WT to increase thrombus formation (Fig. 4f, g), confirming that the MARTX toxin of *V. vulnificus* is mainly responsible for the procoagulant activity of RBCs and thrombus formation.

We also elucidated the biochemical and molecular mechanism underlying the procoagulant activity and morphological changes of the RBCs infected with *V. vulnificus*. *V. vulnificus* infection at the sub-hemolytic level elevates the $[Ca^{2+}]_i$ level with increased caspase-3 and scramblase activities accompanied by PS exposure and MV generation of RBCs (Fig. 2a–e). These molecular alterations suggest that the elevation of the $[Ca^{2+}]_i$ level of the RBCs infected with *V. vulnificus* is key to the procoagulant activity and morphological changes of RBCs, which ultimately enhances the prothrombotic activity (Fig. 2g–i). Although the previous studies reported that the elevation of the $[Ca^{2+}]_i$ level in RBCs is closely linked with thrombotic complications and hemolytic anemia caused by various external stress[24], the elevation of the $[Ca^{2+}]_i$ level in RBCs mediated by a pathogen infection has never been reported to the best of our knowledge.

While purified bacterial toxins listeriolysin O (LLO) and alpha-hemolysin (HlyA) were also observed to induce the elevation of the $[Ca^{2+}]_i$ level in RBCs in vitro[25,26], their effects on the procoagulant and prothrombotic activity and subsequent thrombosis in vivo have not been addressed. Mutational analysis revealed that the MARTX toxin of *V. vulnificus* was responsible for the elevation of the $[Ca^{2+}]_i$ level, leading to the procoagulant activity of RBCs and thrombin generation (Fig. 3b–f). Notably, further functional dissection analysis revealed that the pore-forming activity of the EF-MARTX toxin, but not the multiple cytotoxic or cytopathic effector functions, is attributed to the elevation of the $[Ca^{2+}]_i$ level, which triggers PS exposure of RBCs and thrombin generation (Fig. 3g–j). Previous studies reported that the exposure of RBCs to exogenous and endogenous stimuli like calcium ionophore A23187 and lysophosphatidic acid (LPA) induces a morphological change from discocyte into echinocyte mediated through the elevation of $[Ca^{2+}]_i$ level[11,27]. These results suggest that the pore-forming activity of the EF-MARTX toxin may be responsible for the morphological change of RBCs from normal discocyte into pathologic echinocyte through the elevation of the $[Ca^{2+}]_i$ level.

It has been previously reported that HlyU, a transcription regulator, upregulates the *rtxA* encoding MARTX toxin by directly binding to its promoter[28]. Interestingly, this study revealed that the expression of *rtxA* was significantly increased following the exposure of *V. vulnificus* to RBCs (Fig. 3a and Supplementary Table 2), supporting the previous report that direct contact of *V. vulnificus* to host cells is required to display MARTX cytotoxicity[29]. However, the transcriptomic analysis showed that, unlike *rtxA*, *hlyU* encoding HlyU was not upregulated in *V. vulnificus* infecting RBCs (Supplementary Table 2), indicating that other mechanism(s), yet unknown, could be involved in the upregulation of *rtxA* following exposure of *V. vulnificus* to RBCs. In addition to the *rtxA* gene, several genes related to iron uptake (Supplementary Table 2) were also significantly upregulated in *V. vulnificus* exposed to RBCs, along with the generation of numerous MVs. Recent reports suggest that MVs derived from RBCs retain residual hemoglobin as an important resource of iron[30]. Incidentally, iron is essential for the growth and lethality of *V. vulnificus*[31]. Our results on the upregulation of iron uptake-related genes of *V. vulnificus* combined with the numerous MVs generation from *V. vulnificus*-infected RBCs imply that increased availability of iron from the MVs may promote the growth and lethality of *V. vulnificus*, which may be crucial to the manifestation of its pathogenicity.

*V. vulnificus* infection may accompany circulatory disorders from blood clotting and blockade of the venous vessels leading to deep vein thrombosis[3,5,21,32,33], highlighting the importance of thrombosis for the pathological complications of *V. vulnificus* infection. However, the mechanism underlying *V. vulnificus* infection-associated thrombosis has not been fully elucidated. Here, we investigated the role of RBCs and MARTX toxin in coagulation and thrombosis associated with *V. vulnificus* infection. A pivotal role of RBCs in hemostasis and thrombosis has been well explained in recent reviews[8,9,34]. Several mechanisms were suggested for the involvement of RBCs in the promotion of blood coagulation and thrombosis. Previous studies suggested that RBC aggregation may aggravate deep vein thrombosis by increasing the hydrodynamic resistance in the veins in the lower limbs[9,18]. However, it is yet to be clarified whether *V. vulnificus*-induced RBC aggregation alone is sufficient to trigger venous thrombosis. Attachment of RBCs to vascular endothelium also plays a role in hemostasis and thrombosis. Unlike normal RBCs, RBCs are prone to attach to endothelium under certain pathological conditions such as sickle cell disease[35]. It has been suggested that PS exposure in RBCs was linked to increased adhesion to endothelium[17,36]. Consistently with this notion, our in vitro results demonstrated that PS-exposing RBCs following *V. vulnificus* infection increased endothelial cell attachment (Fig. 2h) and RBC aggregation (Fig. 2i), indicating that these events may promote the vascular occlusions associated with venous thrombosis observed in the *V. vulnificus*-infected patients.

More attention has recently been paid to the role of PS exposure of RBCs in coagulation and thrombosis[8,9,34]. Under pathological states, negatively charged PS in the surface of the RBC membrane provides sites for the assembly of prothrombinase complex with factor Xa and factor Va, which facilitates thrombin generation from prothrombin to enhance blood clotting and ultimately promote venous thrombosis[13]. In accordance with this downstream component of coagulation, our in vitro results showed that the PS-exposing RBCs and MVs occurring upon *V. vulnificus* infection increase thrombin generation by exogenous addition of factor Xa, and factor Va (Fig. 2g), suggesting that PS-exposing RBCs accelerate the downstream component of coagulation cascade to promote coagulation in vivo even though coagulation is not triggered by them. Previous studies reported that *V. vulnificus* infection secretes a metalloprotease to activate factor XII, that involves the first step of the intrinsic pathway[37–39]. In addition, there are several reports that infection with bacteria such as *Staphylococcus aureus* initiates the extrinsic coagulation pathway by inducing tissue factor production from monocytes and endothelium via inflammatory mediators such as TNF-alpha[40–42]. There is no report on the increase of tissue factor expression by *V. vulnificus* infection, but the levels of TNF-alpha were significantly increased in *V. vulnificus*-infected patients[43–45]. It is possible that *V. vulnificus* infection may induce the expression of tissue factors on the surface of cells such as monocytes by the increased TNF-alpha, allowing the complex formation with factor VII, finally activating Factor X. Collectively, *V. vulnificus* infection may induce inflammation and activate the upstream coagulation factors such as tissue factor and factor XII in extrinsic/intrinsic pathway, which culminates in the formation of factors Xa/Va. These factors Xa/Va readily form the prothrombinase complex in the presence of PS-exposing RBCs resulting in increased generation of thrombin from prothrombin, ultimately forming fibrin clot.

In summary, we discovered that *V. vulnificus* infection induces the morphological and functional changes of RBCs, ultimately leading to thrombosis (Fig. 5). This study is the first to demonstrate molecular mechanisms of bacterial infection leading to the thrombotic complications through inducing the procoagulant and prothrombotic activity of RBCs. We believe that our study provides an important insight into understanding pathogen-induced thrombosis at molecular levels, which could help to develop innovative strategies to control the intravascular coagulation caused by pathogens.

# Morphological changes

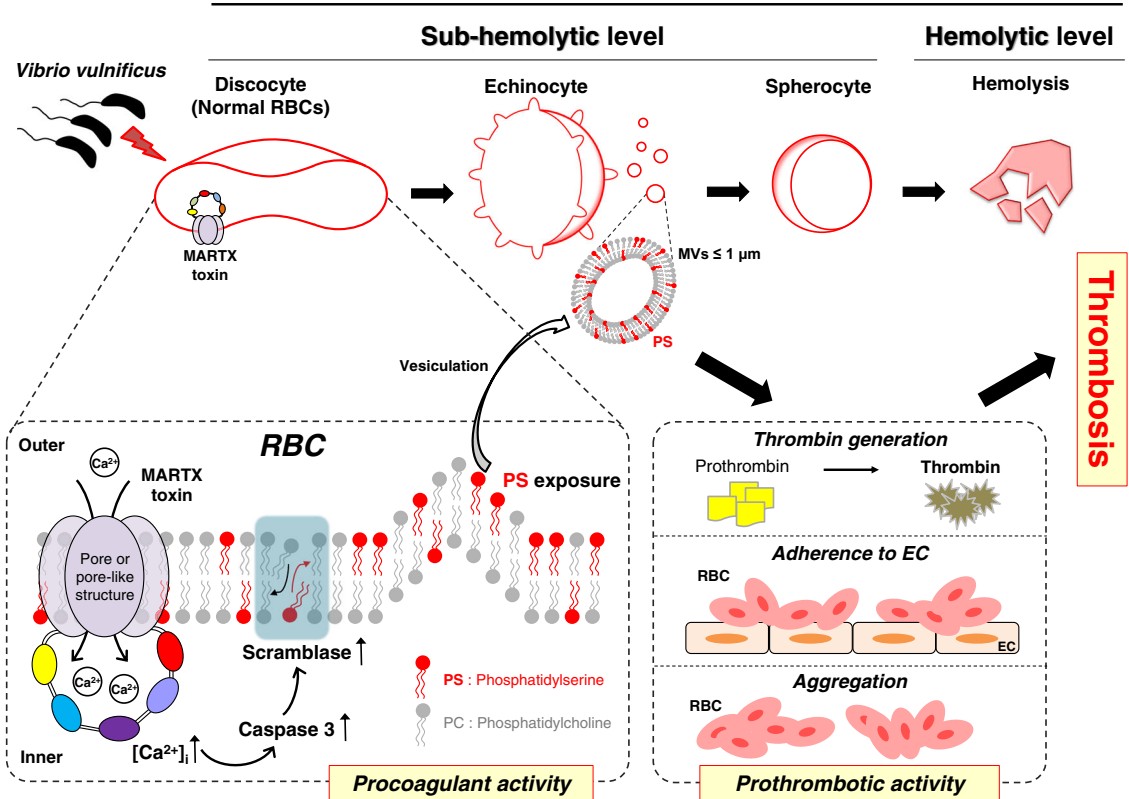

**Fig. 5 | Summary: effects of the *V. vulnificus* MARTX toxin on the procoagulant and prothrombotic activity of human RBCs leading to thrombosis.** *V. vulnificus* infects RBCs at the sub-hemolytic level and produces the pore-forming MARTX toxin that induces procoagulant activity accompanied by the elevation of the $[Ca^{2+}]_i$ level, caspase-3, and scramblase activity, then leading to PS exposure and MV generation. Procoagulant activity is responsible for the shape changes from discocyte to spherocyte, enhancing the prothrombotic activity such as thrombin generation, adherence to EC, and self-aggregation of the RBCs, which ultimately promotes thrombosis. Different colors in the MARTX toxin represent each effector domain. PS phosphatidylserine, MV microvesicle, and EC endothelial cell.

## Methods

### Strains, plasmids, culture conditions, and mutant construction

The strains and plasmids used in this study are listed in Table S1[14,46–50]. A clinical isolate of *V. vulnificus* MO6-24/O[46] and its three isogenic mutants were used in this study. The *ΔrtxA* mutant, which was constructed by deleting 89% of the *rtxA* structural gene and then inserting a *nptI* gene cassette conferring resistance to kanamycin into the deleted *rtxA*, produces no MARTX toxin[14]. The *rtxA::nptI* mutant with the insertion of *nptI* at the 5′-end of *rtxA* also produces no MARTX toxin[47]. The revertant, in which the *rtxA* structural gene was restored from the *rtxA::nptI* mutation, produces the WT-MARTX toxin[47]. To construct the *EF-rtxA* mutant, the pDS_EF-MARTX[50], carrying *rtxA* with the deletion of 34% structural gene encoding effector domains of the MARTX toxin, was conjugally transferred into the *V. vulnificus* WT and a resulting mutant by the double homologous recombination was screened out. Desired mutation and reversion were validated by sequencing the *rtxA* gene. Growths of the *V. vulnificus* strains were obtained in a Luria-Bertani medium supplemented with 2% NaCl (w/v) (LBS) and appropriate antibiotics at 30 °C and monitored by measuring $A_{600}$ of the cultures.

### Preparation of human RBCs

With the approval from the Ethics Committee of the Health Service Center at Seoul National University (IRB No. 1909/001-007), all participants provided written informed consent before enrollment for the study. Human blood was obtained from healthy male donors (20–30 years old) using a vacutainer with acid citrate dextrose (ACD) and a 21-gauge needle (Becton Dickinson) on the day of each experiment. Platelet-rich plasma and buffy coat were removed by aspiration after centrifugation at 200×*g* for 15 min. Packed human RBCs were washed two times with phosphate-buffered saline (PBS; 1.06 mM $KH_2PO_4$, 154 mM NaCl, 2.96 mM, and $Na_2HPO_4$ at pH 7.4) and once more with Ringer's solution (125 mM NaCl, 5 mM KCl, 1 mM $MgSO_4$, 32 mM HEPES, and 5 mM glucose, pH 7.4). The washed RBCs were resuspended in Ringer's solution to a concentration of $1 × 10^8$ cells/ml, and $CaCl_2$ was added to a final concentration of 1 mM prior to use for further experiments.

### Detection of hemolytic activity

The RBCs resuspension infected with *V. vulnificus* at various MOIs and incubated for different incubation times was centrifuged at 10,000×*g* for 2 min. The extent of hemolysis of the RBCs was determined by measuring released hemoglobin spectrophotometrically at 540 nm ($A_{540}$). The complete lysis of RBCs by 1% Triton X-100 (Sigma-Aldrich) was expressed as 100% hemolysis.

### Microscopy analyses

For transmission electron microscopy (TEM) analysis, the RBCs were infected with *V. vulnificus* for 1 h and then fixed with 2% glutaraldehyde solution in the refrigerator overnight. On the second day, the fixed RBCs were washed three times with PBS and then post-fixed with 1% osmium tetroxide for 30 min. The fixed RBCs were washed briefly with

distilled water twice and serially dehydrated with 30, 50, 70, 80, and 90% ethanol, and finally with 100% ethanol three times. Next, the dehydrated RBCs were treated twice with propylene oxide for 10 min each, infiltrated once with the 1:1 mixture of propylene oxide and Spurr's resin for 2 h, and then finally left with only Spurr's resin in a desiccator overnight. On the third day, the RBCs were infiltrated again with fresh Spurr's resin for 2 h in a desiccator and then kept in a 70 °C oven overnight for polymerization of the resin. Finally, the RBCs were analyzed by TEM (JEM-1010, JEOL). For scanning electron microscopy (SEM) analysis, the RBCs were infected with *V. vulnificus* for 1 h and then fixed with 2% glutaraldehyde solution (Sigma-Aldrich) for 1 h at 4 °C. The fixed RBCs were then centrifuged, washed three times with PBS, and then post-fixed with 1% osmium tetroxide (Sigma-Aldrich) for 30 min at room temperature. After washing with PBS twice, the RBCs were dehydrated serially with 50, 70, 80, 90, and 100% ethanol, coated with gold and then analyzed by SEM (Merlin Compact FE-SEM, Zeiss). For confocal observation, the RBCs were diluted to $3 \times 10^6$ cells/ml, added to a 4-well-chambered coverslip (Thermo Fisher), and then incubated at RT for 1 h for complete attachment. Unattached RBCs were removed by washing with Ringer's solution containing 2% BSA, and then infected with *V. vulnificus*. After infection, the RBCs were stained with the phycoerythrin-labeled monoclonal mouse anti-human CD235a antibody (anti-glycophorin-A-PE (BD Bioscience, 555570, 1:100 dilution)) for 30 min, washed with Ringer's solution once, and then analyzed by confocal microscopy equipped with an argon laser (TCS SP8, Leica). Excitation and emission filters were set at 488 and 550–600 nm, respectively. To observe PS exposure, the RBCs were stained with fluorescein-isothiocyanate (FITC)-labeled Annexin V (Annexin V-FITC (BD Bioscience, 556419, 1:50 dilution) instead of glycophorin-A-PE.

## Flow cytometry analysis

PS exposure and MV generation of the RBCs infected with *V. vulnificus* were analyzed using the FACSCalibur flow cytometer (Becton Dickinson) equipped with an argon-ion laser emitting at 488 nm. The Annexin V-FITC (BD Bioscience, 556419, 1:20 dilution) was used to detect PS exposure, whereas the anti-glycophorin-A-PE (BD Bioscience, 555570, 1:20 dilution) was used to determine RBCs and MV generation. RBCs and MV were identified by the relative size of cells (forward-scatter height, FSC-H) and granularity (side-scatter height, SSC-H). Fluorescence data from 5,000 events were collected and analyzed using Cell Quest Pro software v.6.0. When required, the RBCs were pretreated with caspase-3 inhibitors (Z-DEVD-FMK and Q-VD-OPh (Calbiochem)) before being infected with *V. vulnificus*. A gate for PS exposure-positive events was set using the analyzed data from RBCs treated with 2.5 mM of CaCl₂ instead of 2.5 mM EDTA before the flow cytometry analysis. Data from 5000 events were collected and analyzed using Cell Quest Pro software v.6.0. For the detection of the intracellular Ca²⁺ level, the RBCs were pretreated with 3 µM of Fluo-4 acetoxymethyl ester (fluo-4 AM) (Thermo Fisher Scientific) for 1 h at 37 °C in the dark were infected with *V. vulnificus* and then the fluorescence of Fluo-4 AM was analyzed by flow cytometry. For the evaluation of caspase-3 activity, three hundred microliters of the RBCs infected with *V. vulnificus* were mixed with 1 µL of the in situ fluorescent marker for caspase-3 (FITC-DEVD-FMK (Calbiochem)) and then incubated for 30 min in a thermomixer (37 °C, 1000 rpm, dark). Then, the RBCs were centrifuged (1000×*g* for 5 min), washed twice with Ringer's solution, and resuspended in 500 µL of Ringer's solution. The fluorescence from the RBC suspension was analyzed by flow cytometry. Data from 10,000 events were collected and analyzed. For the assessment of scramblase activity, 500 µL of the RBCs were infected with *V. vulnificus* in the presence of 0.5 µL of fluorescent phosphatidylcholine (PC) (C6-NBD-PC (Avanti Polar Lipids)) for 0, 10, and 30 min at 37 °C. Then, 50 µL of aliquots were transferred to tubes containing 450 µL of ice-cold Ringer's solution either with or without 1% BSA and

further incubated for 10 min. Because the PCs associated with the outer leaflet of the RBC membrane would be back-extracted by the BSA, the degree of PC translocation into the inner leaflet of the RBC membrane was measured by comparing the fluorescence intensity of the cells before (without 1% BSA) and after (with 1% BSA) back-extraction. Data from 5000 events were collected and analyzed using Cell Quest Pro software v.6.0. Percentage of PC translocation was calculated by the following: PC translocation (%) = (fluorescence intensity after back-extraction/fluorescence intensity before back-extraction) × 100.

## Prothrombinase assay

The RBCs were infected with *V. vulnificus*, washed with Ringer's solution with 1 mM of CaCl₂, and then incubated with 5 nM of factor Xa (Hematologic Technologies) and 10 nM of factor Va (Hematologic Technologies) in Tyrode buffer (134 mM NaCl, 10 mM HEPES (Sigma-Aldrich), 5 mM glucose, 2.9 mM KCl, 1 mM MgCl₂, 12 mM NaHCO₃, 0.34 mM Na₂HPO₄, 0.3% BSA, and 2 mM CaCl₂ at pH 7.4) for 3 min at 37 °C. After the addition of 2 µM of purified human prothrombin (factor II) (Hematologic Technologies) to the suspension for 3 min, a 10 µl aliquot was transferred to a tube containing 490 µl of stop buffer (50 mM Tris-HCl, 120 mM NaCl, and 2 mM EDTA at pH 7.9). Thrombin generation was determined by measuring $A_{405}$ and 1 µmol thrombin generation per min is defined as a unit (U) per $10^6$ RBCs in vitro and per µL whole blood cells ex vivo[11]. In experiments using purified Annexin V (BD Bioscience, 556416, final concentration 2 uM)[51], human RBCs were pre-incubated with the Annexin V for 10 min, and then infected with *V. vulnificus* at 5, and 10 MOIs for 30 min. Thrombin generation was determined using the prothrombinase assay as mentioned above.

## Adherence and aggregation of RBCs to EC and self-aggregation

Human umbilical vein endothelial cells were originally purchased from Lonza (C2517A). The RBCs pre-infected with *V. vulnificus* for 30 min were moved to human umbilical vein endothelial cells (ECs, $2 \times 10^4$ cells) pre-stained with Calcein green (Invitrogen) and then incubated at 37 °C for 30 min. The RBCs pre-infected but not adhered to ECs were washed out using the EC growth medium (EBM-2 (Lonza)). The RBCs pre-infected and adhered to ECs were stained with the anti-glycophorin-A-PE (BD Bioscience, 555570, 1:100 dilution) and then analyzed by fluorescence microscopy (Axiovert 200 M, Zeiss). To observe self-aggregation, the RBCs were infected with *V. vulnificus*, stained with the anti-glycophorin-A-PE (BD Bioscience, 555570, 1:100 dilution), and then analyzed by fluorescence microscopy.

## Transcriptome analysis

For transcriptome analysis, the freshly prepared RBCs were infected with *V. vulnificus* MO6-24/O WT at an MOI of 10. After 30 min incubation at 37 °C, total RNAs from the *V. vulnificus* were isolated[52]. Libraries of total RNAs were prepared with the TruSeq stranded mRNA library prep kit (Illumina) according to the manufacturer's protocol and then sequenced on the Illumina NovaSeq6000 platform. The sequencing reads were mapped to the complete reference genome of *V. vulnificus* MO6-24/O (GenBank accession numbers: CP002469 and CP002470, https://www.ncbi.nlm.nih.gov/), and the expression level of each gene was calculated as a relative log expression (RLE) value. The RLE values were normalized and analyzed statistically using DESeq2 v.1.26.0 to identify differentially expressed genes ($P < 0.05$) when exposed to the RBCs.

## Animal experiments

All the protocols used for animal experiments were approved by the Ethics Committee of the Animal Service Center at Seoul National University (SNU-170417-27-5). For bridge studies, the rat RBCs were prepared from the whole blood (citrate-dextrose treated)

withdrawn from the abdominal aorta of male Sprague-Dawley (SD) rats (280–350 g, 8 weeks old) anesthetized with urethane (1.25 g/kg, intraperitoneal injection) following the method used for the human RBC preparation, and then, PS exposure and thrombin generation were determined as mentioned above. For ex vivo studies, whole blood cells were collected from the abdominal aorta of the rats after 1 h IV infection with *V. vulnificus* and then, thrombin generation was determined using the prothrombinase assay as mentioned above. For in vivo RBC morphological studies, rats were infected IV with either *V. vulnificus* WT or *ΔrtxA* mutant for 1 h, and then whole blood cells were collected. RBCs were prepared after centrifugation at 5800×*g* for 1 min followed by removal of platelet-rich plasma and buffy coat. RBC morphology was analyzed using SEM as mentioned above. For in vivo thrombus formation studies, a venous thrombosis rat model was used, based upon the previous report that thrombus formed in this model was due to RBCs and fibrin clots with only a few platelets[53]. The rat abdomen was surgically opened, and the inferior vena cava (IVC) was exposed after 1 h IV infection with *V. vulnificus*. Both ends of the IVC, about 16 mm apart from each other, were tied loosely. However, all side branches were tied tightly with cotton threads. Thromboplastin (Instrumentation Laboratory) was infused into the IVC for 1 min to initiate thrombus formation. Upon tightening the cotton threads at the ends of the IVC, the abdominal cavity was provisionally closed to maintain the blood stasis for 15 min. The tied IVC segments were excised and opened longitudinally to isolate the thrombus, of which weights were immediately determined. The thrombus was photographed at a 1× magnification using a Stemi 305 stereomicroscope (Zeiss) equipped with an Axiocam 105 color camera (Zeiss).

### Statistical analysis
The means and standard errors (SE) of means were calculated from at least five independent experiments. The data were subjected to a two-way analysis of variance (ANOVA) followed by Duncan's multiple range test or two-tailed Student's *t*-test to determine statistical significance when compared with the control using SigmaPlot v.12.

### Reporting summary
Further information on research design is available in the Nature Research Reporting Summary linked to this article.

## Data availability
All source data to generate dot plots in this study are available in the source data Excel sheet. Total mRNA sequences from transcriptome analysis were deposited in the NCBI Sequence Read Archive (SRA; http://www.ncbi.nlm.nih.gov/sra) under accession numbers SRR14307536, SRR14307537, SRR14307538, SRR14307539, SRR14307540, and SRR14307541. Source data are provided with this paper.

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

## Acknowledgements

This work was supported by the National Research Foundation of Korea, funded by the Ministry of Science, ICT, and Future Planning (2017R1E1A1A01074639 and 2021K1A3A1A20001134 to S.H.C.; 2021R1I1A1A01049980 to H.Y.C.).

## Author contributions

H.Y.C., B.S.K., and S.H.C. designed the research. H.Y.C. and B.S.K. performed the research. H.Y.C, Y.B., K.-M.L., B.S.K., and S.H.C. analyzed the data and wrote the manuscript. S.H.C supervised the study. All authors have read and approved the final version of the manuscript.

## Competing interests

The authors declare no competing interests.
