## [Peer review file · Nature Communications]

Reviewers' comments:

Reviewer #1 (Remarks to the Author):

Chung H Y et al in their manuscript titled "Vibrio vulnificus MARTX toxin induces thrombosis through the procoagulant activity of red blood cell" examine the effect of low multiplicity of infection [MOI] (1-10 cells) of *V. vulnificus* on human red blood cells (RBCs) in a highly controlled set of experiments. Their experiments revealed that upon exposure of *V. vulnificus* to RBCs, human blood cells induce a series of both intra- and extracellular chemicals, and compounds/molecules that together changes RBC's cell morphology from normal to echinocyte to spherocyte that becomes prone to thrombosis. This is an excellent study with all experiments done with carefully designed experiments. I have the following comments to the manuscript as follows:

1. I suggest that the authors change their current title to "Pore-forming domain of MARTX toxin of *Vibrio vulnificus* induces thrombosis through the procoagulant activity of red blood cells", as effector molecules of the toxin did not play any role(s) on RBCs' phenotypic changes and thrombosis.
2. The manuscript is very difficult to read and understand because of poor writing and composition; therefore, I suggest that the authors extensively revise the manuscript with particular emphasis on abstract and result sections. Many sentences for example in MATRIX toxin (result section) must be revised for clarity!
3. I assume that the authors indicated the expression of phosphatidyl serine (PS) rather than PS exposure in the entire manuscript including all figures; if I am correct, please change accordingly in the manuscript and in figure legends/axis.
4. Do the RBCs self-aggregation linked to PS, as PS promoted the spike like cell surface appendages on RBC? Can the authors use the same dye (annexin V) to illuminate PS in figure 2-i to test the idea I have raised?
5. If effector domain of MARTX does not have any role (as shown experimentally) in the manuscript, does pore-forming domain of the toxin elicits stress on RBCs to render them to spherocyte as reported by mechanical and chemical stressors (should be discussed with references in discussion and result sections)
6. In addition to PS, do extracellular MVs contain other chemicals such as iron and/or any other material that might be of interest to *V. vulnificus*' growth and/or signaling purposes?
7. Humans, particularly immunocompromised patients (liver cirrhosis, diabetes mellitus, cancer), are incidental host of *V. vulnificus*. Given that what is the overall impact of this study in terms of deformed RBCs. Even If human body find such deformed RBCs, isn't that our immune system would remove them? If so, sub-optimal level of infection may not render the infected person to be anemic? Please discuss in the "discussion section"

8. Alternatively, deformed RBCs might go apoptosis to help the affected host from not being anemic?
Discuss in your discussion

Reviewer #2 (Remarks to the Author):

The article by Chung et al present important findings regarding the pathogenesis of the poorly understood pathogen *Vibrio vulnificus*. *V. vulnificus* is a deadly pathogen that can cause a deadly septicemia characterized by circulatory disorders such as venous thrombosis. To date, the molecular mechanisms controlling this phenomenon remain poorly understood. This study aims to understand some of the factors associated with this phenotype.

The study is well designed and systematic. Limited attention has been spent regarding the grammar and narrative of the article (even the first two sentences of the abstract end the same way). Furthermore, the findings are exclusively descriptive and corroborate those found in other pathogens. It would be of interest if the authors were more explicit about the broader relevance of their findings.

Reviewer #3 (Remarks to the Author):

Infection by *Vibrio vulnificus* can lead to life threatening sepsis and related pathological effects such as venous thrombosis. The mechanism by which *V. vulnificus* causes thrombosis had not previously been elucidated and it is the subject of this study. It is known that venous thrombosis can result from increased procoagulant function by red blood cells (RBCs) and that other markers such as elevation of intracellular calcium, caspase-3 and scramblase activities can all occur in response to certain stressors. Additional associated changes of the RBCs include externalization of phosphatidylserine (PS), which increases prothrombotic activity, shedding of PS-bearing microvesicles (MVs) and morphotype switching from the normal discocyte to an echinocyte and then a spherocyte form. Here, the authors present evidence that exposure of RBCs to sub-hemolytic numbers of *V. vulnificus* results in the indicated morphotype changes, production of MVs as well as PS externalization, and procoagulant and prothrombotic activities (Figs. 1 and 2). They then used transcriptome sequencing to identify *V. vulnificus* genes differentially expressed in response to exposure to RBCs. Only one exotoxin gene (*rtxA*), which encodes MARTX toxin, was upregulated. Previous studies have shown that MARTX toxin is the most important exotoxin for pathogenesis of *V. vulnificus*. They then found that an *rtxA* deletion strain showed significantly lower induction of procoagulant activities of RBCs (PS externalization, MV production, other markers) (Fig. 3). They also present evidence that a modified MARTX toxin minus its

effector domains but still containing its pore-forming domains (EF-rtxA) still induces procoagulant activities in RBCs (also Fig. 3). Finally, the authors found that wild type *V. vulnificus* but not an rtxA deletion mutant caused venous thrombosis in vivo in rats. The authors concluded that the MARTX toxin (and specifically its pore-forming ability) is the molecule responsible for inducing thrombosis during *V. vulnificus* infection. Overall, this is a substantial result since it relates an important pathological feature of *V. vulnificus* infection to a specific virulence determinant produced by the bacterium.

Additional comments

1. Line 70- rtxA is mentioned here for the first time without any introduction. It should be stated more clearly that it is the structural gene for the MARTX toxin.

2. Fig 1C- the TEM is not particularly informative. There are red arrowheads pointing to some cells that are *V. vulnificus* but other cells of similar size apparently aren't. But how can we really tell? The SEM and confocal images are much more definitive. Why not delete the TEM or at least put it in supplemental info?

3. Results on lines 150-160. The authors begin using a rtxA deletion mutant to study the role of MARTX toxin in inducing procoagulant/thrombin effects on RBCs in vitro (Fig. 3) and in vivo in rats (Fig. 4). The comparison is the WT strain versus the rtxA deletion mutant. According to supplemental table1, the rtxA gene appears to have been replaced by a resistance gene in this mutant. The authors do not include the results of a complemented mutant in any of their data in these figures. However, they mention in this section that an insertional mutant of rtxA was attenuated (similar to the rtxA deletion mutant) in inducing procoagulant activities in RBCs and the data is in supplemental Fig. 2. Also, the revertant of the insertion mutant was restored to WT for its ability to induce procoagulant activities (also in supplemental Fig. 2). But these address only the vitro results. The demonstration of a lack of thrombosis in rats by the rtxA deletion mutant (Fig. 4) is a critically important piece of this study. As such, at a minimum, either the complemented rtxA deletion mutant should be included in the in vivo experiment or, alternatively, the WT, the insertion mutant and its revertant should be compared.

4. Lines 169-170- please include reference(s) where it was shown that pore forming ability of MARTX toxin or its derivatives is indicated by bleb formation.

5. Discussion section- given the overall impact of the results of this study, the discussion section is rather thin and includes a lot of restating of the results (e.g., most of the third paragraph). In that paragraph, the authors mention that its interesting that the rtxA gene is upregulated following exposure of *V. vulnificus* to RBCs. I agree that its interesting, but can the authors give any more insight/speculation as to the possible mechanism of this induction? What is known about regulation of rtxA gene expression

and could this induction help explain previous evidence which showed that MARTX (RTX) toxin displays contact-mediated cytotoxicity (e.g., Kim et al, 2008)?

Reviewer #4 (Remarks to the Author):

This study shows that MARTX toxin from *Vibrio vulnificus* increase phosphatidylserine (PS) exposure on RBC and the release of extracellular vesicles. Most of the studies are in vitro. In the rat model thrombosis is triggered by tissue factor.

Major comments

1/ An increase in levels of PS-positive RBCs and extracellular vesicles will enhance ongoing coagulation but is not sufficient to trigger coagulation. Therefore, the story is incomplete.

2/ With the exception of sepsis, there is little data supporting the idea that *Vibrio vulnificus* triggers coagulation.

3/ MARTX toxin similar to listeriolysin and alpha hemolysin toxin are designed to lyse RBC. The pathophysiological significance of studying sub-hemolytic doses of MARTX is questionable.

4/ *Vibrio vulnificus* express metalloproteases that activate prothrombin to thrombin. In addition, these proteases cleave fibrinogen but this does not form a clot. Since this is a Gram-negative bacteria it is likely that like *E. coli* and other Gram-negative bacteria the mechanism of activation of coagulation is due to induction of tissue factor expression on monocytes.

5/ The rat studies show enhanced venous thrombosis. This model requires tissue factor as a trigger. What is not shown are levels of circulating PS-positive extracellular vesicles. A model of disseminated intravascular coagulation would be a better model rather than venous thrombosis.

**Reviewer's Comments by Reviewer #1.**

**General comment:**

Chung H Y et al in their manuscript titled “*Vibrio vulnificus* MARTX toxin induces thrombosis
through the procoagulant activity of red blood cell” examine the effect of low multiplicity of
infection [MOI] (1-10 cells) of *V. vulnificus* on human red blood cells (RBCs) in a highly
controlled set of experiments. Their experiments revealed that upon exposure of *V. vulnificus* to
RBCs, human blood cells induce a series of both intra- and extracellular chemicals, and
compounds/molecules that together changes RBC's cell morphology from normal to
echinocyte to spherocyte that becomes prone to thrombosis. This is an excellent study with all
experiments done with carefully designed experiments. I have the following comments to the
manuscript as follows:

**Comment #1:** I suggest that the authors change their current title to “**Pore-forming
domain of MARTX toxin of *Vibrio vulnificus* induces thrombosis through the
procoagulant activity of red blood cells**”, as effector molecules of the toxin did not play
any role(s) on RBCs' phenotypic changes and thrombosis.

**Response:** We really appreciate the reviewer's kind suggestion to change the title which is
more relevant to our work. We modified the title as the reviewer suggested.

**(Original)** *Vibrio vulnificus* MARTX toxin induces thrombosis through the procoagulant
activity of red blood cell

**(Revised)** Pore-forming activity of the MARTX toxin of *Vibrio vulnificus* induces
thrombosis through the procoagulant activity of red blood cells

**Comment #2:** The manuscript is very difficult to read and understand because of poor
writing and composition; therefore, I suggest that the authors extensively revise the
manuscript with particular emphasis on abstract and result sections. Many sentences for
example in MARTX toxin (result section) must be revised for clarity!

**Response:** We have modified the abstract to emphasize the significance of our work. Also
we have clearly stated not to confuse *rtxA* gene with MARTX toxin in results section. The
final text was completely checked by a professional native speaker.

**(Original Abstract)**

The human pathogen *Vibrio vulnificus* is often characterized by hemolysis and circulatory
disorders such as venous thrombosis. However, little is known about the molecular
mechanism and responsible virulence factor(s) of *V. vulnificus* for venous
thrombosis. Herein, *V. vulnificus* infection at the sub-hemolytic level induced shape change
of human red blood cells (RBCs) into echinocytes, followed by spherocytes. Also, *V.*
*vulnificus* induced procoagulant activity of RBCs, leading to phosphatidylserine exposure,

and microvesicle generation and ultimately, prothrombotic activity. Furthermore, the
exposure of *V. vulnificus* to RBCs substantially increased the *rtxA* expression, and the pore-
forming activity of the multifunctional autoprocessing repeats-in-toxin (MARTX) toxin
promoted the elevation of the intracellular Ca^{2+} [Ca^{2+}]_i level. Most importantly, the MARTX
toxin was essential for thrombin generation *ex vivo* and thrombus formation of RBCs in the
rat *in vivo*. Collectively, we demonstrated that the *V. vulnificus* MARTX toxin may induce
thrombosis through the cytopathic changes of RBCs in the infected patients.

**(Revised Abstract)**

*V. vulnificus*-infected patients suffer from hemolytic anemia and circulatory lesions, often
accompanied by venous thrombosis. However, the pathophysiology of venous thrombosis
associated with *V. vulnificus* infection remains largely unknown. Herein, *V. vulnificus*
infection at the sub-hemolytic level induced shape change of human red blood cells (RBCs)
accompanied by phosphatidylserine exposure, and microvesicle generation, leading to the
procoagulant activation of RBCs and ultimately, acquisition of prothrombotic activity. Of
note, *V. vulnificus* exposed to RBCs substantially upregulated the *rtxA* gene encoding
multifunctional autoprocessing repeats-in-toxin (MARTX) toxin. Mutant studies showed
that *V. vulnificus*-induced RBC procoagulant activity was due to the pore forming region of
the MARTX toxin causing intracellular Ca^{2+} influx in RBCs. In a rat venous thrombosis
model, the *V. vulnificus* wild type increased thrombosis while the *ΔrtxA* mutant failed to
increase thrombosis, confirming that *V. vulnificus* induces thrombosis through the
procoagulant activation of RBCs via the mediation of the MARTX toxin.

**Comment #3: I assume that the authors indicated the expression of phosphatidyl serine**
**(PS) rather than PS exposure in the entire manuscript including all figures; if I am correct,**
**please change accordingly in the manuscript and in figure legends/axis.**

**Response:** If cellular protein levels were changed, it is usually called the decrease or
increase of its expression. Since PS is an anionic phospholipid mainly localized on the inner
leaflet of cell membranes, it is more relevant to be called 'PS exposure', meaning
externalization of PS from inner membrane into outer membrane. Most references selected
the terms 'PS exposure' or 'PS externalization' rather than 'PS expression' (Nagata S, et al.
Curr Opin Immunol. 2020 62:31-38).

**Comment #4: Do the RBCs self-aggregation linked to PS, as PS promoted the spike like**
**cell surface appendages on RBC? Can the authors use the same dye (annexin V) to**
**illuminate PS in figure 2-i to test the idea I have raised?**

**Response:** It is very interesting question, but it is not feasible to observe RBCs self-
aggregation using the dye (Annexin V-FITC) which was applied to detect the increased PS
exposure on the surface of RBC membranes (Fig. 2b). Since Annexin V is a specific PS-
binding protein (Yen TC, et al. Anal. Biochem. 2010 406(1):70-79), addition of annexin V
would cover the exposed PS and affect RBC self-aggregation. Thus, RBCs were directly
stained with a general dye, anti-glycophorin-A-PE to visualize RBC self-aggregation (Fig.
2i).

**Comments #5: If effector domain of MARTX does not have any role (as shown**
**experimentally) in the manuscript, does pore-forming domain of the toxin elicits stress**
**on RBCs to render them to spherocyte as reported by mechanical and chemical stressors**
**(should be discussed with references in discussion and result sections)**

**Response:** As the reviewer suggested, we have added some information in the 3rd paragraph
of Discussion section describing the possible role of pore-forming regions related to
echinocyte formation with a couple of references.

**(Original, the 3rd paragraph in Discussion section)**

Subsequently, mutational analysis demonstrated that the MARTX toxin was responsible for
the elevation of the $[Ca^{2+}]_i$ level leading to the procoagulant activity of RBCs with thrombin
generation (Fig. 3b-f). Notably, further functional dissection analysis revealed that the pore-
forming activity of the EF-MARTX toxin, not the multiple cytotoxic or cytopathic effector
functions, is attributed to the elevation of the $[Ca^{2+}]_i$ level, which triggers PS exposure and
thrombin generation of RBCs (Fig. 3g-j).

**(Revised, the 3rd paragraph in Discussion section)**

Mutational analysis revealed that the MARTX toxin was responsible for the elevation of the
$[Ca^{2+}]_i$ level leading to the procoagulant activity of RBCs and thrombin generation (Fig. 3b-
f). Notably, further functional dissection analysis revealed that the pore-forming activity of
the EF-MARTX toxin, but not the multiple cytotoxic or cytopathic effector functions, is
attributed to the elevation of the $[Ca^{2+}]_i$ level, which triggers PS exposure of RBCs and
thrombin generation (Fig. 3g-j). Previous studies reported that the exposure of RBCs to
exogenous and endogenous stimuli like calcium ionophore A23187 and lysophosphatidic
acid (LPA) induces morphological change from discocyte into echinocytes mediated
through the elevation of $[Ca^{2+}]_i$ level (Anderson RA, et al. *Nature*, 1981 **292**(5819):158-161;
Chung SM, et al. *Arterioscler Thromb Vasc Biol*. 2007 **27**(2):414-421). These results
suggest that the pore-forming activity of the EF-MARTX toxin may be responsible for the
morphological change of RBCs from normal discocyte into pathologic echinocyte through
the elevation of the $[Ca^{2+}]_i$ level.

**Comments #6: In addition to PS, do extracellular MVs contain other chemicals such as**
**iron and/or any other material that might be of interest to *V. vulnificus*' growth and/or**
**signaling purposes?**

**Response:** Reviewer indicated a very important hypothesis on the possible role of MVs
formed by *V. vulnificus*-infected RBCs. Recent reports suggest that MVs derived from RBCs
retain residual hemoglobins and metabolic proteins (Chiangjong W, et al. *Front Med*. 2021
**8**:761362) and that iron may play a key role in the pathogenicity of *V. vulnificus* (Wen Y, et
al. *J Biol Chem*. 2016 **291**(27):14213-14230). As shown in supplementary table 2, *V.*
*vulnificus* infection of RBCs significantly upregulated several genes of *V. vulnificus* related
to iron uptake, suggesting that increased availability of iron from numerous MVs may
promote the growth and survival of *V. vulnificus*.

We have added a new discussion paragraph with a couple of references right after the 4th
paragraph of Discussion section as follows.

**(Revised, new addition in Line 242)**

In addition to the *rtxA* gene, several genes related to iron uptake (Supplementary Table 2)
were also significantly upregulated in *V. vulnificus* exposed to RBCs, along with the
generation of numerous MVs. Recent reports suggest that MVs derived from RBCs retain
residual hemoglobin as an important resource of iron (Chiangiong W, et al. *Front Med.* 2021
**8:761362**). Incidentally, iron is essential for the growth and lethality by *V. vulnificus* (Wen
Y, et al. *J Biol Chem.* 2016 **291(27):14213-14230**). Our results on the upregulation of iron-
uptake related genes of *V. vulnificus* combined with the numerous MVs generation from *V.*
*vulnificus*-infected RBCs imply that increased availability of iron from the MVs may
promote the growth and lethality of *V. vulnificus*, which may be crucial to the manifestation
of its pathogenicity.

**Comments #7: Humans, particularly immunocompromised patients (liver cirrhosis,**
**diabetes mellites, cancer), are incidental host of *V. vulnificus*. Given that what is the**
**overall impact of this study in terms of deformed RBCs. Even If human body find such**
**deformed RBCs, isn't that our immune system would remove them? If so, sub-optimal**
**level of infection may not render the infected person to be anemic? Please discuss in the**
**“discussion section”**

**Response:** Prior to answer comment #7 and #8, we would like to clarify the terminology
‘deformed RBCs’. Normal RBCs in human are continuously deformed when they pass
through a narrow capillary, helping effective O₂ transport. The echinocytes, and spherocytes
induced by pathogen stimuli in our study would be better called RBC abnormality, since the
ability of O₂ transport was significantly decreased in these abnormal forms of RBCs (Jung
F, et al. *Clinl Hemorheol Microcirc.* 2008 **38:1-11**). Our results show that this RBC
abnormality along with PS exposure and MV generation induced procoagulant activity,
ultimately leading to thrombosis.

As the reviewer suggests, it has been well known that the immunocompromised patients
(liver cirrhosis, diabetes mellitus, and cancer) have an increased risk of thrombosis and
cardiovascular diseases. There is, however, no epidemiological study available on the risk
of thrombosis in *V vulnificus* infected immunocompromised patients.

Regarding the reviewer’s question on the removal of abnormal RBCs, the abnormal RBCs
exposing PS on the surface of membrane could exhibit the dual effect; procoagulant activity
and erythrophagocytosis (taken up by the spleen) (Klei TR, et al. *Front Immunol.* 2017 **8:**
**73**). The controlling mechanism of procoagulant activity vs erythrophagocytosis is currently
unknown. However, since pathogen infection is quite a rapid response inducing PS exposure
and MV generation within an hour, we think the procoagulant pathway would be
predominant compared to erythrophagocytosis.

At this moment, due to a lack of experiment on procoagulant activity using *in vitro* cell
system simulating immunocompromised patients, no report on an increased risk by

epidemiological study, and absence of mechanistic support for rapid vs slow responses, we
are reluctant to discuss this part, which may require too much speculation. We hope the
reviewer will agree with our view.

**Comments #8: Alternatively, deformed RBCs might go apoptosis to help the affected host**
**from not being anemic? Discuss in your discussion**

**Response:** This question may be answered in part in the response to Comments #7. The *V.*
*vulnificus* infection of RBCs was a rapid process while apoptosis is a somewhat delayed and
latent response. At sub-hemolytic concentrations of *V. vulnificus*, the infection of RBCs
rapidly induced attachment of RBCs to endothelial cells or aggregation of RBCs (Fig. 2h,
i), resulting in the decreased number of intact RBCs. Also at high concentrations, *V.*
*vulnificus* caused rapid morphological changes, finally leading to hemolysis (Fig. 5). In both
cases, the deformed RBCs may rapidly cause anemia in the affected host depending upon
time or severity of infection.

**Reviewer's Comments by Reviewer #2**

**General comment:**

The article by Chung et al present important findings regarding the pathogenesis of the poorly
understood pathogen *Vibrio vulnificus*. *V. vulnificus* is a deadly pathogen that can cause a
deadly septicemia characterized by circulatory disorders such as venous thrombosis. To date,
the molecular mechanisms controlling this phenomenon remain poorly understood. This study
aims to understand some of the factors associated with this phenotype.

The study is well designed and systematic. Limited attention has been spent regarding the
grammar and narrative of the article (even the first two sentences of the abstract end the same
way). Furthermore, the findings are exclusively descriptive and corroborate those found in
other pathogens. It would be of interest if the authors were more explicit about the broader
relevance of their findings.

**Response #1:** Thank you for your favorable response on our work. We have modified the
abstract especially the first two sentences and the last sentence to increase the clarity of our
work. The final text was completely checked by a professional native speaker.

**(Original Abstract)**

The human pathogen *Vibrio vulnificus* is often characterized by hemolysis and circulatory
disorders such as venous thrombosis. However, little is known about the molecular
mechanism and responsible virulence factor(s) of *V. vulnificus* for venous
thrombosis. Herein, *V. vulnificus* infection at the sub-hemolytic level induced shape change
of human red blood cells (RBCs) into echinocytes, followed by spherocytes. Also, *V.*
*vulnificus* induced procoagulant activity of RBCs, leading to phosphatidylserine exposure,
and microvesicle generation and ultimately, prothrombotic activity. Furthermore, the
exposure of *V. vulnificus* to RBCs substantially increased the *rtxA* expression, and the pore-
forming activity of the multifunctional autoprocessing repeats-in-toxin (MARTX) toxin
promoted the elevation of the intracellular Ca^{2+} level. Most importantly, the MARTX toxin
was essential for thrombin generation *ex vivo* and thrombus formation of RBCs in the rat *in*
*vivo*. Collectively, we demonstrated that the *V. vulnificus* MARTX toxin may induce
thrombosis through the cytopathic changes of RBCs in the infected patients.

**(Revised Abstract)**

*V. vulnificus*-infected patients suffer from hemolytic anemia and circulatory lesions, often
accompanied by venous thrombosis. However, the pathophysiology of venous thrombosis
associated with *V. vulnificus* infection remains largely unknown. Herein, *V. vulnificus*
infection at the sub-hemolytic level induced shape change of human red blood cells (RBCs)
accompanied by phosphatidylserine exposure, and microvesicle generation, leading to the
procoagulant activation of RBCs and ultimately, acquisition of prothrombotic activity. Of
note, *V. vulnificus* exposed to RBCs substantially upregulated the *rtxA* gene encoding
multifunctional autoprocessing repeats-in-toxin (MARTX) toxin. Mutant studies showed
that *V. vulnificus*-induced RBC procoagulant activity was due to the pore forming region of
the MARTX toxin causing intracellular Ca^{2+} influx in RBCs. In a rat venous thrombosis
model, the *V. vulnificus* wild type increased thrombosis while the Δ *rtxA* mutant failed to

increase thrombosis, confirming that *V. vulnificus* induces thrombosis through the
procoagulant activation of RBCs via the mediation of the MARTX toxin.

**Response #2:** Regarding the comment that our findings may be an isolated event which
cannot be extended to other pathogens, it has been reported that many pathogens such as *E.*
*coli* and *Salmonella* also cause thrombosis (Farstad H, et al. *Acta paediatrica*. 2003
92(2):254-257; Pineda MC, et al. *Neurologist*. 2012 18(4):202-203). However, they didn't
provide molecular mechanisms for the pathogens to cause the thrombosis. Therefore, we
can suggest that the increased procoagulant activity accompanied by calcium influx into
RBCs can explain the thrombosis caused by other pathogens such as *E. coli* and *Salmonella*
in addition to *Vibrio* spp. carrying *rtxA* homologues. In order to extend our findings to other
pathogens, we are currently investigating whether other pathogens may induce the
procoagulant activity of RBCs and what type of toxin is responsible for the thrombotic
events observed in human patients. We would like to report the further results with other
pathogens as another independent paper in near future.

**Reviewer's Comments by Reviewer #3**

**General comment:**

Infection by *Vibrio vulnificus* can lead to life threatening sepsis and related pathological effects
such as venous thrombosis. The mechanism by which *V. vulnificus* causes thrombosis had not
previously been elucidated and it is the subject of this study. It is known that venous thrombosis
can result from increased procoagulant function by red blood cells (RBCs) and that other
markers such as elevation of intracellular calcium, caspase-3 and scramblase activities can all
occur in response to certain stressors. Additional associated changes of the RBCs include
externalization of phosphatidylserine (PS), which increases prothrombotic activity, shedding of
PS-bearing microvesicles (MVs) and morphotype switching from the normal discocyte to an
echinocyte and then a spherocyte form. Here, the authors present evidence that exposure of
RBCs to sub-hemolytic numbers of *V. vulnificus* results in the indicated morphotype changes,
production of MVs as well as PS externalization, and procoagulant and prothrombotic activities
(Figs. 1 and 2). They then used transcriptome sequencing to identify *V. vulnificus* genes
differentially expressed in response to exposure to RBCs. Only one exotoxin gene (*rtxA*), which
encodes MARTX toxin, was upregulated. Previous studies have shown that MARTX toxin is
the most important exotoxin for pathogenesis of *V. vulnificus*. They then found that an *rtxA*
deletion strain showed significantly lower induction of procoagulant activities of RBCs (PS
externalization, MV production, other markers) (Fig. 3). They also present evidence that a
modified MARTX toxin minus its effector domains but still containing its pore-forming
domains (EF-*rtxA*) still induces procoagulant activities in RBCs (also Fig. 3). Finally, the
authors found that wild type *V. vulnificus* but not an *rtxA* deletion mutant caused venous
thrombosis *in vivo* in rats. The authors concluded that the MARTX toxin (and specifically its
pore-forming ability) is the molecule responsible for inducing thrombosis during *V. vulnificus*
infection. Overall, this is a substantial result since it relates an important pathological feature
of *V. vulnificus* infection to a specific virulence determinant produced by the bacterium.

**Comments #1: Line 70- *rtxA* is mentioned here for the first time without any introduction.**
**It should be stated more clearly that it is the structural gene for the MARTX toxin.**

**Response:** Thank you for your kind notification. In order to clarify the *rtxA*, we have revised
line 57 in introduction section as follows.

**(Original, Line 57)** Among multiple virulence factors of *V. vulnificus*, the multifunctional
autoprocessing repeats-in-toxin (MARTX) toxin is the most crucial exotoxin responsible for
the pathogenicity of *V. vulnificus*^{14,15}.

**(Revised, Line 58)** Among multiple virulence factors of *V. vulnificus*, the multifunctional
autoprocessing repeats-in-toxin (MARTX) toxin, which is encoded by the *rtxA* gene, is the
most crucial exotoxin responsible for the pathogenicity of *V. vulnificus*^{14,15}.

**Comments #2:** Fig 1C- the TEM is not particularly informative. There are red
arrowheads pointing to some cells that are *V. vulnificus* but other cells of similar size
apparently aren't. But how can we really tell? The SEM and confocal images are much
more definitive. Why not delete the TEM or at least put it in supplemental info?

**Response:** We are in agreement with the reviewer's opinion. We will move the TEM data
from Fig. 1 into Supplementary Fig. 2, since some readers might be interesting to look up
the TEM data to compare the images from SEM or Confocal microscopy.

**Comments #3:** Results on lines 150-160. The authors begin using a *rtxA* deletion mutant
to study the role of MARTX toxin in inducing procoagulant/thrombin effects on RBCs in
vitro (Fig. 3) and *in vivo* in rats (Fig. 4). The comparison is the WT strain versus the *rtxA*
deletion mutant. According to supplemental table1, the *rtxA* gene appears to have been
replaced by a resistance gene in this mutant. The authors do not include the results of a
complemented mutant in any of their data in these figures. However, they mention in this
section that an insertional mutant of *rtxA* was attenuated (similar to the *rtxA* deletion
mutant) in inducing procoagulant activities in RBCs and the data is in supplemental Fig.
2. Also, the revertant of the insertion mutant was restored to WT for its ability to induce
procoagulant activities (also in supplemental Fig. 2). But these address only the *vitro*
results. The demonstration of a lack of thrombosis in rats by the *rtxA* deletion mutant
(Fig. 4) is a critically important piece of this study. As such, at a minimum, either the
complemented *rtxA* deletion mutant should be included in the *in vivo* experiment or,
alternatively, the WT, the insertion mutant and its revertant should be compared.

**Response:** We appreciate the reviewer's comment which is very critical to strengthen our
manuscript. As reviewer #3 suggested, we did an additional experiment with a revertant of
the *rtxA* deletion mutant to compare *V. vulnificus* wild type and *rtxA* deletion mutant to rule
out other artefacts for *in vitro* and *in vivo* experiments.

As shown in Fig. 4e, f below, the reversion of the *rtxA* deletion mutant restored the ability
of *V. vulnificus* to increase thrombogenesis along with procoagulant activities of RBCs, as
consistently with *in vitro* data (Supplementary Fig. 3b, c). These results strongly support our
conclusion that MARTX toxin of *V. vulnificus* is mainly responsible for the procoagulant
activity of RBCs and thrombus formation *in vivo*.

We have replaced the original Fig. 4e, f with the new one and revised the last paragraph of
results section, accordingly.

(Original Fig. 4e, f)

(Revised Fig. 4e, f)

(Original, the last paragraph of Results section)

Finally, effects of *V. vulnificus* infection on thrombus formation *in vivo* were examined using
a rat venous thrombosis model (Fig. 4d). When rats were IV infected with the *V. vulnificus*
WT, and then injected with the thromboplastin, thrombus formation increased in an

incubation time-, and an infectious dose-dependent manner (Fig. 4e, f). In contrast, rats
infected with *ΔrtxA* mutant did not show any significant increase in thrombus formation
when compared with the uninfected control, indicating that the MARTX toxin is essential
for the thrombosis *in vivo* induced by *V. vulnificus* infection.

**(Revised, the last paragraph of Results section)**

Finally, effects of *V. vulnificus* infection on thrombus formation were examined *in vivo* using
a rat venous thrombosis model (Fig. 4d). When rats were infected with the *V. vulnificus* WT
through IV, and then injected with the thromboplastin, thrombus formation increased in an
incubation time-, and an infectious dose-dependent manner (Fig. 4e, f). In contrast, rats
infected with the *ΔrtxA* mutant did not show any significant increase in thrombus formation
when compared with the uninfected control. Furthermore, the revertant of the *ΔrtxA* mutant
rescued the ability of *V. vulnificus* to increase thrombus formation, as consistent with *in vitro*
data (Supplementary Fig. 3b, c). These results support our conclusion that the MARTX toxin
of *V. vulnificus* is mainly responsible for the procoagulant activity of RBCs and thrombus
formation *in vivo*.

**Comments #4: Lines 169-170- please include reference(s) where it was shown that pore**
**forming ability of MARTX toxin or its derivatives is indicated by bleb formation.**

**Response:** Thank you for your kind suggestion. We will include a reference investigating
pore forming ability of MARTX toxin related to bleb formation in line 171. (Cho C, et al. *J*
*Microbiol.* 2022 **60**(2):224-233)

**Comments #5: Discussion section- given the overall impact of the results of this study, the**
**discussion section is rather thin and includes a lot of restating of the results (e.g., most of**
**the third paragraph). In that paragraph, the authors mention that its interesting that the**
***rtxA* gene is upregulated following exposure of *V. vulnificus* to RBCs. I agree that its**
**interesting, but can the authors give any more insight/speculation as to the possible**
**mechanism of this induction? What is known about regulation of *rtxA* gene expression**
**and could this induction help explain previous evidence which showed that MARTX**
**(RTX) toxin displays contact-mediated cytotoxicity (e.g., Kim et al, 2008)?**

**Response:** As the reviewer suggested, we have augmented the 3rd paragraph of discussion
section emphasizing the mechanism of *rtxA* gene expression and its significance in RBCs
thrombosis following *V. vulnificus* infection. We have included several additional citations
to support our view.

**(Revised, new addition in Line 234)**

It has been previously reported that HlyU, a transcription regulator, upregulates the *rtxA*
encoding MARTX toxin by directly binding to its promoter (Lee ZW, et al. *mBio.* 2020
**11**(4):e00723-00720). Interestingly, this study revealed that the expression of *rtxA* was
significantly increased following the exposure of *V. vulnificus* to RBCs (Fig. 3a,

Supplementary Table 2), supporting the previous report that direct contact of *V. vulnificus* to
host cells is required to display MARTX cytotoxicity (Kim YR, et al. *Cell Microbiol.* 2008
10(4):848–862). However, transcriptomic analysis showed that unlike *rtxA*, *hlyU* encoding
HlyU was not upregulated in *V. vulnificus* infecting RBCs (Supplementary Table 2),
indicating that other mechanism(s), yet unknown, could be involved in the upregulation of
*rtxA* following exposure of *V. vulnificus* to RBCs.

**Reviewer's Comments by Reviewer #4**

**General comments:**

This study shows that MARTX toxin from *Vibrio vulnificus* increase phosphatidylserine (PS)
exposure on RBCs and the release of extracellular vesicles. Most of the studies are *in vitro*. In
the rat model thrombosis is triggered by tissue factor.

**Response to general comments:** In order to resolve the reviewer #4 concern, we have
performed several experiments to support our views and answered the questions on a
point-by-point basis. Furthermore, one paragraph of discussion section was incorporated
to augment the significance of our work.

**Major comments**

**Comments #1: An increase in levels of PS-positive RBCs and extracellular vesicles will**
**enhance ongoing coagulation but is not sufficient to trigger coagulation. Therefore, the**
**story is incomplete.**

**Response:** The reviewer #4 questioned the pivotal role of PS-exposure of RBCs and RBCs-
shed extracellular vesicles in initiating blood coagulation.

It has been well-established that PS-exposing RBCs and extracellular microvesicles actively
and majorly participate in the coagulation and thrombosis in various pathological conditions.
Especially, a pivotal role of RBCs in thrombosis has been well and reiteratively explained
in recent reviews published in major journals in hematology (Byrnes JR, et al. *Blood*. 2017
130(16):1795-1799; Matthew F. et al. *Blood*. 2012 120(18):3837-3845; Weisel JW, et al. *J*
*Thromb Haemost.* 2019 17(2):271-282). These reviews highlight that PS exposure of RBCs
can induce procoagulant activity, which is directly linked with the promotion of thrombosis.
In line with this, our results showed that *V. vulnificus*-infected RBCs increases thrombin
generation by the exogenous addition with factor Xa, factor Va and factor II (prothrombin)
in a MOI-, and time-dependent manner (Fig 2g), indicating that PS-exposing RBCs can
trigger thrombin generation, ultimately leading to coagulation and thrombosis.

Furthermore, it is well-established that the impairment of procoagulant-anticoagulant
balance can lead to thrombosis in various clinical settings (van Baal, et al. *Thromb*
*Haemost.* 2000 83(01):9-34; Stuijver, DJ, et al. *Thromb. Haemost.* 2012 108(6):077-1088),
indicating that *V. vulnificus* infection-induced RBC PS exposure can trigger thrombin
generation in the blood, ultimately leading to clinically important thrombus formation. This
is fully confirmed by our *in vivo* study where normal healthy rats infected with *V. vulnificus*
showed increased thrombus formation, while those infected with *rtxA* deletion mutant didn't
(Fig. 4e, f). Besides, even in the absence of blood coagulation factors, PS-exposing RBCs
show increased endothelial cell attachment (Fig. 2h) and RBC aggregation (Fig. 2i),
indicating that *V. vulnificus* infection-induced RBC PS exposure can explain various
hematologic complications of *V. vulnificus* infection.

**Comments #2: With the exception of sepsis, there is little data supporting the idea that**
***Vibrio vulnificus* triggers coagulation.**

**Response:** We understand that the reviewer raised the question regarding the clinical
evidence of our study to *V. vulnificus* infection. Indeed, clinical case reports directly linked
with coagulation were mainly about sepsis following *V. vulnificus* infection. However, there
were case reports about the clinical symptoms related with the circulatory disorders from
blood clotting and blockade of venous vessel such as deep vein thrombosis (You JS, et al.
*Am J Emerg Med.* 2012 **30**(9):2098.e5-6; Torres L, et al. *Eur J Clin Microbiol Infect Dis*
2002, **21**:537-538; Hong GL, et al. *Burns.* 2012 **38**(2):290-295; Matsuoka Y., et al. *Braz J*
*Infect Dis.* 2013 **17**(1):7-12; Mascola, L., et al. CDC MMWR. 1996 **45**:621-624), supporting
the clinical relevance of our study.

In addition, there are some reports that many people with *V. vulnificus* infection require the
surgery of lower limb amputation (Tsai YH, et al. *J Trauma*, 2009 **66**(3):899-905; Matsuoka
Y., et al. *Braz J Infect Dis.* 2013 **17**(1):7-12), which is mainly caused by the blockade of
peripheral blood vessels resulting in the necrosis of the affected tissue. It has been also reported
that lower limb amputation in hospital is frequently associated with deep vein thrombosis,
suggesting that thrombosis might be an important pathogenicity of *V. vulnificus* (Burke B, et
al. *Am J Phys Med Rehabil.* 2000 **79**(2):145-149; Yeager RA, et al. *J Vasc Surg.* 1995
**22**(5):612-615). We consider that these reports sufficiently support that *V. vulnificus* triggers
coagulation.

**Comments #3: MARTX toxin similar to listeriolysin and alpha hemolysin toxin are**
**designed to lyse RBC. The pathophysiological significance of studying sub-hemolytic**
**doses of MARTX is questionable.**

**Response:** Previous studies demonstrated that hemolysis is not an abrupt but progressively
occurring event. Even though *bona fide* hemolysis, i.e., the rupture of RBCs is not shown at
sub-hemolytic doses, significant changes in RBC membranes can precede such as
morphological and biochemical perturbations as we demonstrated in our study (Fig. 1c, e).
Even at this sub-hemolytic doses, RBCs exposed PS and shed PS-bearing microvesicles,
which could actively promote coagulation pathways (Fig. 2a, g). Although, at hemolytic
doses, these phenomena would get much stronger and severer via the additional generation
of PS-bearing RBC debris (Bian Y, et al. *Food Chem Toxicol*, 2019 **131**:110553), the
pathophysiological features of RBCs observed with sub-hemolytic doses of *V. vulnificus*
may provide more essential clues to understand the progression of its pathogenesis.

**Comments #4: *Vibrio vulnificus* express metalloproteases that activate prothrombin to**
**thrombin. In addition, these proteases cleave fibrinogen but this does not form a clot.**
**Since this is a Gram-negative bacteria it is likely that like *E. coli* and other Gram-**
**negative bacteria the mechanism of activation of coagulation is due to induction of**
**tissue factor expression on monocytes.**

**Response:** The reviewer raised another interesting hypothesis to explain the increased
coagulation in *V. vulnificus* infection.

It is a significant point which deserves additional paragraph of discussion. Chang et al.
reported that *V. vulnificus* can secrete an extracellular metalloprotease which can produce
thrombin and lyse fibrin (Chang AK, et al. *J Bacteriol.* 2005 **187**(20):6909-16). Incidentally,
these authors stated that the thrombin activating effect of *V. vulnificus* was only transient
and temporary. We understand that in addition to the involvement of these proteases, there
may be other unknown factors contributing to the *V. vulnificus*-induced coagulation. So, to
rule out factors other than MARTX toxin-induced procoagulant activity of RBC, we
compared *V. vulnificus* wild type and *rtxA* deletion mutant of *V. vulnificus* in our *in vitro*
and *in vivo* experiments (Fig. 3b, c, f and Fig. 4f). The *rtxA* deletion mutant of *V. vulnificus*
only induced minimal level of thrombosis *in vivo*. Furthermore, an additional experiment
was performed with a revertant of the *rtxA* deletion mutant as indicated red dot.

**Fig. 4f** Original submission (left) and a new data with an additional experiment with the
revertant of the *rtxA* deletion mutant (right)

The revertant of the *rtxA* deletion mutant restored the ability of *V. vulnificus* to increase
thrombogenesis along with the procoagulant activities of RBCs, as consistently with *in vitro*
data (Supplementary Fig. 3b, c). These results support our conclusion that MARTX toxin of
*V. vulnificus* is mainly responsible for the procoagulant activity of RBCs and thrombus
formation *in vivo*.

Regarding the tissue factor issue raised by reviewer #4, it is possible that the synergistic
effect *in vivo* might occur by the increased tissue factor expression. Since the study on these
synergy needs a series of experiments beyond the scope of the current study, we would like
to investigate further in another independent study. We hope that the reviewer could agree
with our view.

**(Revised, New paragraph in Discussion section)**

*V. vulnificus* infection accompanies the circulatory disorders from blood clotting and

blockade of venous vessel such as deep vein thrombosis (You JS, et al. *Am J Emerg Med.*
2012 **30**(9):2098.e5-6; Torres L, et al. *Eur J Clin Microbiol Infect Dis.*2002 **21**:537-538;
Hong GL, et al. *Burns.* 2012 **38**(2):290-295; Matsuoka Y., et al. *Braz J Infect Dis.* 2013
**17**(1):7-12; Mascola, L., et al. CDC MMWR. 1996 **45**:621-624), highlighting the
importance of thrombosis for the pathological complications of *V. vulnificus* infection.
However, the mechanism underlying *V. vulnificus* infection-associated thrombosis has not
been fully elucidated. Thrombosis is a complex multifactorial process, which involves
various blood coagulation factors and cells other than RBCs. Chang et al. reported that *V.*
*vulnificus* can secrete an extracellular metalloprotease, which can produce thrombin and
lyse fibrin (Chang AK, et al. *J Bacteriol.* 2005 **187**(20):6909-16). However, they indicated
that the thrombin activating effect of *V. vulnificus* was only transient or temporary. Here, we
focused on the role of RBCs and MARTX toxin in coagulation and thrombosis associated
with *V. vulnificus* infection. A pivotal role of RBCs in thrombosis has been well explained
in recent reviews published in major journals in hematology (Byrnes JR, et al. *Blood.* 2017
**130**(16):1795-1799; Matthew F. et al. *Blood.* 2012 **120**(18):3837-3845; Weisel JW, et al. *J*
*Thromb Haemost.* 2019 **17**(2):271-282). These reviews highlight that PS exposure of RBCs
can induce procoagulant activity. In accordance with this, our results showed that the *V.*
*vulnificus*-infected RBCs increase thrombin generation by the exogenous addition of factor
Xa, factor Va and factor II (prothrombin) in an MOI-dependent and a time-dependent
manner (Fig 2g), indicating that the PS-exposing RBCs can trigger thrombin generation,
ultimately leading to coagulation and thrombosis.

Furthermore, the impairment of procoagulant-anticoagulant balance can lead to thrombosis
in various clinical settings (van Baal, et al. *Thromb Haemost.* 2000 **83**(01), 29-34; Stuijver,
DJ, et al. *Thromb Haemost.* 2012 **108**(6):1077-1088), indicating that the *V. vulnificus*
infection-induced RBC PS exposure can trigger thrombin generation in the blood, ultimately
leading to clinically important thrombus formation. This was fully demonstrated by our *in*
*vivo* study where normal healthy rats infected with *V. vulnificus* showed increased PS
exposure of RBCs and thrombus formation while those infected with the *rtxA* deletion
mutant did not show an increase. Furthermore, even in the absence of blood coagulation
factors, the PS-exposing RBCs show increased endothelial cell attachment (Fig. 2h) and
RBC aggregation (Fig. 2i), indicating that the *V. vulnificus* infection-induced RBC PS
exposure can explain various hematologic complications of *V. vulnificus* infection.
Moreover, the revertant of the *rtxA* deletion mutant rescued the ability of *V. vulnificus*
to increase thrombus formation (Fig. 4e, f), confirming that the MARTX toxin of *V. vulnificus*
is mainly responsible for the procoagulant activity of RBCs and thrombus formation.

**Comments #5:** The rat studies show enhanced venous thrombosis. This model requires
tissue factor as a trigger. What is not show are levels of circulating PS-positive
extracellular vesicles. A model of disseminated intravascular coagulation would be a
better model rather than venous thrombosis.

**Response:** Our goal is to investigate the role of RBCs in coagulation and thrombosis.
Therefore, venous thrombosis animal model is the most appropriate to evaluate the role of
*V. vulnificus*-infected RBCs on the increased thrombotic risk *in vivo* since the thrombus
formed in this model is mainly composed of RBCs and fibrin clots with only a few platelets
(Vogel GM, et al. *Thromb Res.* 1989 **54**(5):399-410). We have provided this additional

information in line 368 of Methods Section as below.

**(Original, Line 368)** For *in vivo* studies, the rat abdomen was surgically opened, ----

**(Revised, Line 417)** For *in vivo* studies, a venous thrombosis rat model was used, based
upon the previous report that thrombus formed in this model was due to RBCs and fibrin
clots with only a few platelets (Vogel GM, et al. *Thromb Res.* 1989 **54**(5):399-410). The rat
abdomen was surgically opened, -----

As the reviewer pointed out, in our venous thrombosis rat model, rats were infected with *V.*
*vulnificus* and then infused with tissue thromboplastin in order to expedite the thrombus
formation according to the original method paper (Berry CN, et al. *Br J Pharmacol.* 1994
**113**(4):1209-1214; Bian Y, et al. *Part Fibre Toxicol.* 2021 **18**(1):28; Peternel, L, et al.
*Thromb Res.* 2005 **115**(6):527-534). The reviewer #4 suggested a model of disseminated
intravascular coagulation (DIC) using TF-induced system to examine whether *V. vulnificus*
triggers coagulation cascade. However, DIC is known to address the dysregulation of
processes in coagulation and fibrinolysis including thrombocytopenia, elevated D-dimer
concentration, decreased fibrinogen concentration, and alteration of clotting times. However,
DIC model provides only limited information on RBC abnormalities, which deterred the
adoption of this model in our study.

Regarding the absence of data on the PS-positive extracellular vesicles *in vivo*, unlike *in*
*vitro* experiments, there are many factors interfering the assay to detect PS positive vesicles
in whole blood system *in vivo*. Furthermore, it is hard to identify the tissue origin of vesicles.
To clarify these issues, we performed another experiment to observe the morphological
change of RBCs *in vivo*. One hour after iv injection of *V. vulnificus* wild type and *rtxA*
deletion mutant to rats, blood was collected to observe RBCs using scanning electron
microscopy. Representative image was shown below for review #4. Consistent with *in vitro*
results (Fig. 1c), administration of wild type induced morphological changes of RBCs into
echinocytes, implying that microvesicles were secreted from RBCs *in vivo* (Allan D, et al.
*Nature*, 1976 **261**(5555):58-60). On the other hand, injection of *rtxA* deletion mutant did not
change the shape compared to the untreated control.

**Fig.** *In vivo* morphological changes of RBCs following injection with
*V. vulnificus* wild type (middle) and *rtxA* deletion mutant (right) to rats

To further address the reviewer's comment, we have performed another experiment, which
showed that *V. vulnificus* infection can induce MV generation in rat RBCs *in vitro*. Fig. 4a

has been replaced with the new one bearing MV generation (red dot box) as shown on the
right.

**Fig. 4a** RBCs PS exposure (left) and PS-bearing MV generation in rat RBCs (right)

Collectively, we think that the understanding of pathophysiology of *V. vulnificus* infection
is key to diagnosis and treat the patients infected with *V. vulnificus*. For this, we believe that
our study provides important and novel mechanistic data and various markers which can be
of use to develop effective diagnosis tool following up the prognosis of *V. vulnificus* infected
patients.

We hope that the reviewer #4 could be satisfied with our responses.

REVIEWER COMMENTS

Reviewer #3 (Remarks to the Author):

The authors responses to my comments were mostly reasonable and satisfactory. However, my third comment dealt with the in vivo thrombosis results in Fig. 4. I had stated that either the complemented mutant should be included or the WT, insertion mutant and revertant should be compared. The authors' response was that they included "a revertant of the rtxA deletion mutant", which doesn't make sense. Based on their previous papers, this mutant is deleted for most of the rtxA gene so the full rtxA gene cannot be restored by reversion. It appears the authors are using the revertant derived from their insertion mutant (with derivation as shown in Suppl. Fig. 3). In their revised description of the results for Fig. 4 (lines 193-196) they state "In contrast, rats infected with the (delta)rtxA mutant did not show any significant increase in thrombus formation when compared with the uninfected control. Furthermore, the revertant of the (delta)rtxA mutant rescued the ability of *V. vulnificus* to increase thrombus formation..." Again, this description is incorrect. I think it's likely that the authors' overall conclusions are correct in this paper. However, the in vivo experiment needs to be done correctly. If they're going to show the results of the rtxA deletion mutant then the complemented mutant needs to be included to confirm that defective in vivo thrombus formation by this mutant is due to its rtxA deletion per se. Alternatively, they can show results for the rtxA insertion mutant and its revertant.

Reviewer #4 (Remarks to the Author):

The authors have performed additional experiments that have improved the manuscript. The data (line 537 on the rebuttal) on RBC morphology in RBC isolated from control rats and rats infected with WT and mutant bacteria is important and should be included in the manuscript.

Major comments

1/ The authors are not experts in coagulation and are misinterpreting their data and the literature. Their response to my comments reveals that lack of understanding of coagulation. They also refer to case reports as evidence for thrombosis in patients infected with *Vibrio vulnificus*. This is a rather weak response to my concern that there is very little literature showing that these patients exhibit a high rate of venous thrombosis.

2/ The authors are correct that there are some diseases, such as sickle cell disease, where this is an association between increased PS exposure on RBC and thrombosis/vascular congestion. However, importantly, PS exposure on RBCs alone is not sufficient to initiate coagulation. An increase in PS-exposure on RBCs or other cells, such as platelets, enhances coagulation but does not trigger coagulation. There are 2 ways to trigger coagulation. First, exposure of tissue factor in the vessel wall (hemostasis) or induction of tissue factor on circulating cells, such as monocytes (pathological activation

of coagulation and thrombosis). This is called the extrinsic pathway whereby the tissue factor/factor VIIa complex activates factor X and factor IX. The second pathway is the intrinsic pathway and is driven by the activation of factor XII that then activates factor XI and then this activates FIX. Bacterial infections can lead to induction of tissue factor expression on monocytes and/or activation of factor XII by long chain polyphosphates.

The authors add exogenous factor Xa, factor Va and prothrombin to infected RBCs exposing PS and conclude that the PS-positive RBCs are triggering coagulation. This is not correct. Where did the factor Xa come from in vivo? It must come from either the extrinsic or intrinsic pathway.

The authors nicely demonstrate that *Vibrio vulnificus* infection increases PS exposure on RBC and this is procoagulant. However, they must put this observation into context with what is known about coagulation and add a discussion that they are studying a downstream component of blood coagulation and not the upstream component. Without this paragraph the study's full significance will not be appreciated by the coagulation community.

Reviewer's Comment by Reviewer #3

The authors responses to my comments were mostly reasonable and satisfactory.
However, my third comment dealt with the *in vivo* thrombosis results in Fig. 4. I had
stated that either the complemented mutant should be included or the WT, insertion
mutant and revertant should be compared. The authors' response was that they included
"a revertant of the *rtxA* deletion mutant", which doesn't make sense. Based on their
previous papers, this mutant is deleted for most of the *rtxA* gene so the full *rtxA* gene
cannot be restored by reversion. It appears the authors are using the revertant derived
from their insertion mutant (with derivation as shown in Suppl. Fig. 3). In their revised
description of the results for Fig. 4 (lines 193-196) they state "In contrast, rats infected
with the (Δ)*rtxA* mutant did not show any significant increase in thrombus formation
when compared with the uninfected control. Furthermore, the revertant of the (Δ)
*rtxA* mutant rescued the ability of *V. vulnificus* to increase thrombus formation..." Again,
this description is incorrect. I think it's likely that the authors' overall conclusions are
correct in this paper. However, the *in vivo* experiment needs to be done correctly. If
they're going to show the results of the *rtxA* deletion mutant then the complemented
mutant needs to be included to confirm that defective *in vivo* thrombus formation by this
mutant is due to its *rtxA* deletion per se. Alternatively, they can show results for the *rtxA*
insertion mutant and its revertant.

**Response:** We really appreciate the reviewer's comments to point out that the *rtxA* insertion
mutant (*rtxA::nptI*) should have been used rather than the deletion mutant (Δ *rtxA*) in *in vivo*
thrombosis experiments (Fig. 4f, g). As the reviewer indicated, an *in vivo* experiment using
*rtxA* insertion mutant (*rtxA::nptI*) was performed in order to compare the thrombus
formation of the revertant made from the insertion mutant, which confirmed our conclusion.
Fig. 4f, g has been modified as shown in the next page.

Also, the text describing the results of Fig. 4f, g has been modified as follows,

**(Original, the last paragraph of Results section)**

In contrast, rats infected with the Δ *rtxA* mutant did not show any significant increase in
thrombus formation when compared with the uninfected control. Furthermore, the revertant
of the Δ *rtxA* mutant rescued the ability of *V. vulnificus* to increase thrombus formation, as
consistent with *in vitro* data (Supplementary Fig. 4b, c).

**(Revised, the last paragraph of Results section)**

In contrast, rats infected with the deletion mutant (Δ *rtxA*) as well as the insertion mutant
(*rtxA::nptI*) did not show any significant increase in thrombus formation when compared
with the uninfected control. Furthermore, the revertant of the insertion mutant (*rtxA::nptI*)
rescued the ability of *V. vulnificus* WT to increase thrombus formation (Fig. 4f, g), as
consistent with *in vitro* data (Supplementary Fig. 4b, c).

(Original Fig. 4f)

(Revised Fig. 4f)

38

39 **Fig. 4f** Original thrombus images *in vivo* (left) and a new data from an additional
40 experiment with the *rtxA* insertion mutant (*rtxA::nptI*) (right)

41

(Original Fig. 4g)

(Revised Fig. 4g)

**Fig. 4g** Original figure (left) and a new data from an additional experiment with
the *rtxA* insertion mutant (*rtxA::nptI*) (right)

Reviewer's Comments by Reviewer #4

**General comments:**

**The authors have performed additional experiments that have improved the manuscript.**
 **The data (line 537 on the rebuttal) on RBC morphology in RBC isolated from control rats**
 **and rats infected with WT and mutant bacteria is important and should be included in**
 **the manuscript.**

**Response to general comments:** When we made the first revision, we intended to perform
 the independent complete study on 'e.g. The role of *rtxA* on morphological change of RBCs
 infected with *V. vulnificus*'. We, however, agree with the reviewer's point that this *in vivo*
 data provide an important piece of information to demonstrate the consistency with *in vitro*
 morphological observations (Fig. 1c, e) and to support the role of RBCs *in vivo*. Now, we
 have included the *in vivo* RBC morphological data into Fig. 4d and added the text in the
 Results section as follows,

(Revised, the last paragraph of Results section)

In order to observe the morphological change of RBCs following *V. vulnificus* infection *in vivo*, blood was collected 1 h after IV injection of WT and *ArtxA* mutant to rats.
Representative image of RBCs was observed under scanning electron microscopy (Fig. 4d).
Consistently with *in vitro* results (Fig. 1c, e), administration of WT induced morphological
changes of RBCs into echinocytes (white arrows), implying that PS-bearing microvesicles
were formed from RBCs *in vivo* following *V. vulnificus* infection. On the other hand,
injection of the *ArtxA* mutant did not change the shape compared to the untreated control.

**Comments #1:** The authors are not experts in coagulation and are misinterpreting their
data and the literature. Their response to my comments reveals that lack of
understanding of coagulation. They also refer to case reports as evidence for thrombosis
in patients infected with *Vibrio vulnificus*. This is a rather weak response to my concern
that there is very little literature showing that these patients exhibit a high rate of venous
thrombosis.

**Response:** Regarding the coagulation issue mentioned in the first two sentences, we will
respond in the following comments #2. Even though we provided some references on case
reports of venous thrombosis in the previous revision, we understand the reviewer's concern
that there are not enough evidences for venous thrombosis in *V. vulnificus*-infected patients.
Since *V. vulnificus* infection can lead an acute and rapidly fatal disease with wide clinical
spectrum of symptoms, standard duplex ultrasound imaging test may not be applicable for
diagnosis of deep vein thrombosis.

We, however, have searched for the clinical literature on *V. vulnificus*-infected patients in
order to provide indirect evidences supporting *V. vulnificus*-infected patients may suffer a
significant rate of venous thrombosis.

1) The levels of D-dimer in blood were clinically used to diagnose deep vein thrombosis,
pulmonary embolism, and disseminated intravascular coagulation, etc. Most references
showed that D-dimer level in *V. vulnificus*-infected patients were much higher than a
normal range (<400 ng/mL) (Choi HJ, et al. *J Dermatol.* 2005 **32**(1):48-51; Horseman
MA, et al. *Int J Infect Dis.* 2011 **15**(3):e157–e166; Yu W, et al. *Int J Infect Dis.* 2017
**59**:1-6). For example, Yu et al. reported that D-dimer levels of 6 patients with *V.*
*vulnificus* infection were far beyond the normal range (465, 1293, 9541, 8394, 1841, and
946 ng/mL, respectively), indicating that the patients experienced a blood clotting
condition.

2) Matsuoka et al. reported 12 cases study infected with *V. vulnificus* in Japan showing that
all patients exhibited fever, redness, swelling and sharp pain in the lower limbs at the
first medical examination which symptom is closely related to deep vein thrombosis
(Matsuoka Y, et al. *Braz J Infect Dis.* 2013 **17**(1):7-12). Seven of 12 patients were dead
and 5 patients survived underwent amputation surgery. Another study showed that lower
limbs and lower extremities were obvious location of clinical symptoms in all six
patients infected with *V. vulnificus* (Yu W, et al. *Int. J Infect Dis.* 2017 **59**:1-6), in
accordance with the lower extremity for most common site of deep vein thrombosis

(Ouriel K, et al. *J Vasc Surg.* 2000 **31**(5):895-900). In addition, it has been known that
lower limb amputation in hospital is frequently associated with deep vein thrombosis
(Yeager RA, et al. *J Vasc Surg.* 1995 **22**(5):612-615; Burke B, et al. *Am J Phys Med*
*Rehabil.* 2000 **79**(2):145-149), suggesting that deep vein thrombosis might be an important
pathogenicity of *V. vulnificus*.

3) *V. vulnificus* infection causes several distinct syndromes including primary septicemia,
gastroenteritis, necrotizing fasciitis, and cellulitis (Hendren N, et al. *BMJ Case Rep.*
*2017 bcr2017220199*). Many literature stated that more than one half of patients develop
the typical skin lesions of cellulitis which most commonly occurs in the lower leg (Oliver
JD. *Epidemiol Infect.* 2005 **133**(3):383-391; Bross MH, et al. *Am Fam Physician.*
*2007 76*(4):539-544). Cellulitis shares several clinical features with deep vein
thrombosis that makes very difficult to distinguish between cellulitis and deep vein
thrombosis (Maze MJ, et al. *BMC Infect Dis.* 2013 **13**:141). Furthermore, Bersier et al.
suggests a controversial association between cellulitis and deep vein thrombosis (Bersier
D, et al. *J Thromb Haemost.* 2003 **1**(4):867-868). Comparing the number of literature
describing the clinical feature of cellulitis and deep vein thrombosis, we speculate there
may be a high rate of mis-diagnosis between cellulitis and deep vein thrombosis in local
hospital.

Our study demonstrate that *V. vulnificus* infection could alter RBCs morphology and
increase PS-exposing RBCs and MVs inducing the procoagulant activity, ultimately leading
to venous thrombosis in animal model. Although deep vein thrombosis has not been known
as one of the major clinical symptoms of *V. vulnificus* infection, our mechanistic study *in*
*vitro* as well as *in vivo* may draw clinician's attention in the future to find the clue to diagnose
and treat *V. vulnificus* infected patients.

We hope the reviewer will be satisfied with our response.

**Comments #2:** The authors are correct that there are some diseases, such as sickle cell
disease, where this is an association between increased PS exposure on RBC and
thrombosis/vascular congestion. However, importantly, PS exposure on RBCs alone is not
sufficient to initiate coagulation. An increase in PS-exposure on RBCs or other cells, such
as platelets, enhances coagulation but does not trigger coagulation. There are 2 ways to
trigger coagulation. First, exposure of tissue factor in the vessel wall (hemostasis) or
induction of tissue factor on circulating cells, such as monocytes (pathological activation
of coagulation and thrombosis). This is called the extrinsic pathway whereby the tissue
factor/factor VIIa complex activates factor X and factor IX. The second pathway is the
intrinsic pathway and is driven by the activation of factor XII that then activates factor
XI and then this activates FIX. Bacterial infections can lead to induction of tissue factor
expression on monocytes and/or activation of factor XII by long chain polyphosphates.

The authors add exogenous factor Xa, factor Va and prothrombin to infected RBCs
exposing PS and conclude that the PS-positive RBCs are triggering coagulation. This is
not correct. Where did the factor Xa come from *in vivo*? It must come from either the
extrinsic or intrinsic pathway.

**The authors nicely demonstrate that *Vibrio vulnificus* infection increases PS exposure on**
**RBC and this is procoagulant. However, they must put this observation into context with**
**what is known about coagulation and add a discussion that they are studying a**
**downstream component of blood coagulation and not the upstream component. Without**
**this paragraph the study's full significance will not be appreciated by the coagulation**
**community.**

**Response:** We acknowledge that we may not be an expert in coagulation cascade pathway
and thus, we toned down the discussion related to the coagulation cascade overall.
Furthermore, we misunderstood the previous comments of reviewer #4 a little. Now, we
could understand clearly what exactly the reviewer concerned, which we think is very
important.

We agree that *V. vulnificus* induced RBC PS exposure did not trigger coagulation by itself,
rather it promoted the primed coagulation cascade ultimately leading to increased
thrombosis. To further clarify the role of *V. vulnificus* induced RBC PS exposure in the
coagulation, we performed an additional *in vitro* experiment. As shown below, when the *V.*
*vulnificus* induced PS in RBCs was neutralized by adding the purified Annexin V, thrombin
generation was significantly reduced, suggesting that *V. vulnificus* induced RBC PS
exposure is important for blood coagulation associated with *V. vulnificus* infection.

**Fig.** Effect of purified Annexin V on thrombin generation in *V. vulnificus*-infected RBCs

Since some readers may find this data interesting, we have added it into Supplementary Fig.
3 and modified the text in the Results and Methods section as follows,

**(Revised in Results Section)**

**When the exposed PS in RBCs was blocked with purified Annexin V, the procoagulant**
**activity of *V. vulnificus* was attenuated significantly, suggesting that PS exposure plays a key**
**role in *V. vulnificus*-induced procoagulant activity (Supplementary Fig. 3).**

**(Revised in Methods section)**

In experiments using purified Annexin V (BD Bioscience) (Bae ON, et al. *Chem Res Toxicol.*
2007 10:1760-1768), human RBCs were pre-incubated with the Annexin V (final
concentration 3.5 μ M) for 10 min, and then infected with *V. vulnificus* at 5, and 10 MOIs for
30 min. Thrombin generation was determined using the prothrombinase assay as mentioned
above.

Since two paragraphs in Discussion section which was incorporated in the previous revision
following the reviewer #4's comments was not quite relevant to provide the whole picture
of our work into coagulation and thrombosis, we have dissected and reorganized the
structure of paragraph emphasizing the reviewer's 2nd comments.

**(Two previous paragraphs in Discussion section)**

*V. vulnificus* infection accompanies the circulatory disorders from blood clotting and
blockade of venous vessel such as deep vein thrombosis^{3,5,21,32,33}, highlighting the
importance of thrombosis for the pathological complications of *V. vulnificus* infection.
However, the mechanism underlying *V. vulnificus* infection-associated thrombosis has not
been fully elucidated. Thrombosis is a complex multifactorial process, which involves
various blood coagulation factors and cells other than RBCs. Chang et al. reported that *V.*
*vulnificus* can secrete an extracellular metalloprotease, which can produce thrombin and lyse
fibrin³⁴. However, they indicated that the thrombin activating effect of *V. vulnificus* was only
transient or temporary. Here, we focused on the role of RBCs and MARTX toxin in
coagulation and thrombosis associated with *V. vulnificus* infection. A pivotal role of RBCs
in thrombosis has been well explained in recent reviews published in major journals in
hematology^{8,9,35}. These reviews highlight that PS exposure of RBCs can induce
procoagulant activity. In accordance with this, our results showed that the *V. vulnificus*-
infected RBCs increase thrombin generation by the exogenous addition of factor Xa, factor
Va and factor II (prothrombin) in an MOI-dependent and a time-dependent manner (Fig 2g),
indicating that the PS-exposing RBCs can trigger thrombin generation, ultimately leading
to coagulation and thrombosis.

Furthermore, the impairment of procoagulant-anticoagulant balance can lead to thrombosis
in various clinical settings^{36,37}, indicating that the *V. vulnificus* infection-induced RBC PS
exposure can trigger thrombin generation in the blood, ultimately leading to clinically
important thrombus formation. This was fully demonstrated by our *in vivo* study where
normal healthy rats infected with *V. vulnificus* showed increased PS exposure of RBCs and
thrombus formation while those infected with the *rtxA* deletion mutant did not show an
increase. Furthermore, even in the absence of blood coagulation factors, the PS-exposing
RBCs show increased endothelial cell attachment (Fig. 2h) and RBC aggregation (Fig. 2i),
indicating that the *V. vulnificus* infection-induced RBC PS exposure can explain various
hematologic complications of *V. vulnificus* infection. Moreover, the revertant of the *rtxA*

deletion mutant rescued the ability of *V. vulnificus* to increase thrombus formation (Fig. 4e,
f), confirming that the MARTX toxin of *V. vulnificus* is mainly responsible for the
procoagulant activity of RBCs and thrombus formation (Move to the 1st paragraph in the
Discussion section).

**(Revised, Discussion section)**

*V. vulnificus* infection may accompany the circulatory disorders from blood clotting and
blockade of venous vessel leading to deep vein thrombosis^{3,5,21,32,33}, highlighting the
importance of thrombosis for the pathological complications of *V. vulnificus* infection.
However, the mechanism underlying *V. vulnificus* infection-associated thrombosis has not
been fully elucidated. Here, we investigated the role of RBCs and MARTX toxin in
coagulation and thrombosis associated with *V. vulnificus* infection. A pivotal role of RBCs
in hemostasis and thrombosis has been well explained in recent reviews^{8,9,34}. Several
mechanisms were suggested the involvement of RBCs in the promotion of blood coagulation
and thrombosis. RBC aggregation itself promotes deep venous thrombosis by increasing the
hydrodynamic resistance in blood vessels with low shear, such as the veins in the lower
limbs (Yu F, et al. *J Thromb Haemost.* 2011 **9**(3):481-488). In addition to RBC aggregation,
attachment of RBCs to vascular endothelium also plays a role in hemostasis and thrombosis.
Unlike normal RBCs, RBCs are prone to attach to endothelium under certain pathological
conditions such as sickle cell disease (Frenette PS, et al. *J Clin Invest.* 2007 **117**(4):850-858).
It has been suggested that PS exposure in RBCs was linked to an increased adhesion to
endothelium (Yang Y, et al. *J Biol Chem.* 2010 **285**(52):40489-40495; Wautier MP, et al. *J*
*Thromb Haemost.* 2011 **9**(5):1049-1055). Consistently with this notion, our *in vitro* results
demonstrated that PS-exposing RBCs following *V. vulnificus* infection increased endothelial
cell attachment (Fig. 2h) and RBC aggregation (Fig. 2i), indicating that these events may
promote the vascular occlusions associated with venous thrombosis observed in the *V.*
*vulnificus*-infected patients.

More attention has recently been paid to the role of PS exposure of RBCs in coagulation and
thrombosis^{8,9,34}. Under pathological states, negatively charged PS in the surface of RBC
membrane provides sites for the assembly of prothrombinase complex with factor Xa and
factor Va, which facilitates thrombin generation from prothrombin to enhance blood clotting
and ultimately, promote venous thrombosis¹³. In accordance with this downstream
component of coagulation, our *in vitro* results showed that the PS-exposing RBCs and MVs
occurring upon *V. vulnificus* infection increase thrombin generation by exogenous addition
of factor Xa, and factor Va (Fig. 2g), suggesting that PS-exposing RBCs accelerate the
downstream component of coagulation cascade to promote coagulation *in vivo* even though
coagulation is not triggered by them. Previous studies reported that *V. vulnificus* infection
secretes a metalloprotease to activate factor XII, that involves in the first step of the intrinsic
pathway (Miyoshi S, et al. *Toxicon.* 2004 **44**(8):887-893; Frick IM, et al. *Thromb Haemost.*
**2007 98**(3):497-502; Park JE, et al. *Biochem Biophys Res Commun.* 2014 **450**(2):1099-
**1103**). In addition, there are several reports that infection with bacteria such as
*Staphylococcus aureus* initiates the extrinsic coagulation pathway by inducing tissue factor
production from monocytes and endothelium via inflammatory mediators such as TNF-

alpha (Darke TA, et al. *J Infect Dis.* 1988 **157**(4):749-756; Lowy FD. *N Engl J Med.* 1998
**339**(8):520-532; Lopes-Bezerra LM, et al. *Braz J Med Biol Res.* 2003 **36**(8):987-991). There
is no report on the increase of tissue factor expression by *V. vulnificus* infection, but the
levels of TNF-alpha were significantly increased in *V. vulnificus*-infected patients (Powell
JL, et al. *Infect Immun*, 1997 **65**(9):3713-3718; Shin SH, et al. *FEMS Immunol Med*
*Microbiol.* 2002 **33**(2):133-138; Lee NY, et al. *Mol Immunol.* 2011 **49**(1-2):143-154). It is
possible that *V. vulnificus* infection may induce the expression of tissue factor on the surface
of cells such as monocytes by the increased TNF-alpha allowing the complex formation with
factor VII, finally activating Factor X. Collectively, *V. vulnificus* infection may induce
inflammation and activate the upstream coagulation factors such as tissue factor and factor
XII in extrinsic/intrinsic pathway, which culminates in the formation of factors Xa/Va. These
factors Xa/Va readily form the prothrombinase complex in the presence of PS-exposing
RBCs resulting in increased generation of thrombin from prothrombin, ultimately forming
fibrin clot.

We honestly hope that our responses will be enough to satisfy the reviewer.

REVIEWER COMMENTS

Reviewer #3 (Remarks to the Author):

The authors have included the results of the rtxA insertion mutant in the in vivo thrombosis experiment shown in Fig. 4. The results now clearly indicate that MARTX toxin is primarily responsible for this pathology associated with *Vibrio vulnificus* infection.

Reviewer #4 (Remarks to the Author):

Chung et al Review

Comments.

1/ The title is misleading. MARTX toxin does not “induce” thrombosis through the procoagulant activity of red blood cells. Rather, MARTX toxin induces the exposure of PS on the surface of red blood cells that contributes to thrombosis. The trigger for thrombosis is not studied.

2/ In the abstract, the authors should change “In a rat venous thrombosis model,” to “In a rat venous thrombosis model triggered by tissue factor and stasis, “.

3/ Line 276. “RBC aggregation itself promotes deep vein thrombosis ... 18”. Currently, it is unclear if RBC contribute to venous thrombosis or are simply passively trapped in the clots. I do not believe that RBC aggregation alone is sufficient to induce venous thrombosis. Sickle cell disease is complex and vascular occlusion by RBC and leukocyte aggregates rather than venous thrombosis caused by a clot.

4/ Comment in rebuttal letter. “The levels of D-dimer in blood were clinically used to diagnose deep vein thrombosis, pulmonary embolism, and disseminated intravascular coagulation, etc. Most references 86 showed that D-dimer level in *V. vulnificus*-infected patients were much higher than a 87 normal range. The authors are incorrect. Levels of D-dimer are NOT used to diagnose deep vein thrombosis. Rather, a low level of D-dimer is used to EXCLUDE venous thrombosis.

**Reviewer's Comments by Reviewer #4**

**Comments #1:** The title is misleading. MARTX toxin does not “induce” thrombosis
through the procoagulant activity of red blood cells. Rather, MARTX toxin induces the
exposure of PS on the surface of red blood cells that contributes to thrombosis. The
trigger for thrombosis was not studied.

**Response:** We believe the reviewer #4 suggested more appropriate title to describe our
research clearly. We have previously modified the title adding the phrase ‘Pore-forming
activity of ---’ as the reviewer #1 suggested. Due to the maximum title length is 15 word,
we have changed the title as follows.

**(Original Title)** “Pore-forming activity of the MARTX toxin of *Vibrio vulnificus* induces
thrombosis through the procoagulant activity of red blood cells”

**(Revised Title)** “MARTX toxin of *Vibrio vulnificus* induces RBC phosphatidylserine
exposure that can contribute to thrombosis”

**Comments #2:** In the abstract, the authors should change “In a rat venous thrombosis
model,” to “In a rat venous thrombosis model triggered by tissue factor and stasis”.

**Response:** As the reviewer suggests to describe the animal model more clearly, we have
modified the text of line 26 in Abstract as follows.

**(Original Abstract)** “In a rat venous thrombosis model, ---”

**(Revised Abstract)** “In a rat venous thrombosis model triggered by tissue factor and stasis,
---”

**Comments #3:** Line 276. “RBC aggregation itself promotes deep vein thrombosis ... 18”.
Currently, it is unclear if RBC contribute to venous thrombosis or are simply passively
trapped in the clots. I do not believe that RBC aggregation alone is sufficient to induce
venous thrombosis. Sickle cell disease is complex and vascular occlusion by RBC and
leukocyte aggregates rather than venous thrombosis caused by a clot.

**Response:** We agree with the reviewer’s comments that RBC aggregation alone may not
be sufficient to induce venous thrombosis. Thus, we have modified the text to tone down
the role of increased RBC aggregation in triggering venous thrombosis, as follows,

**(Original in Discussion section)** RBC aggregation itself promotes deep venous
thrombosis by increasing the hydrodynamic resistance in blood vessels with low shear,
such as the veins in the lower limbs¹⁸. In addition to RBC aggregation, attachment of RBCs
to vascular endothelium also plays a role in hemostasis and thrombosis.

(Revised in Discussion section) Previous studies suggested that RBC aggregation may
aggravate deep vein thrombosis by increasing the hydrodynamic resistance in the veins in
the lower limbs^{9,18}. However, it is yet to be clarified whether *V. vulnificus*-induced RBC
aggregation alone is sufficient to trigger venous thrombosis. Attachment of RBCs to
vascular endothelium also plays a role in hemostasis and thrombosis.

**Comments #4: Comment in rebuttal letter. “The levels of D-dimer in blood were clinically**
**used to diagnose deep vein thrombosis, pulmonary embolism, and disseminated**
**intravascular coagulation, etc. Most references showed that D-dimer level in *V.***
***vulnificus*-infected patients were much higher than a normal range. The authors are**
**incorrect. Levels of D-dimer are NOT used to diagnose deep vein thrombosis. Rather, a**
**low level of D-dimer is used to EXCLUDE venous thrombosis.**

**Response:** We really sorry for our incorrect statement in the previous rebuttal letter.

As the reviewer mentioned, a positive or elevated D-dimer levels may suggest a blood
clotting condition, but it can't discriminate whether the type of clotting condition is DVT,
PE, DIC or stroke etc. On the other hand, low or normal D-dimer levels can exclude the
clotting disorder.

We mistakenly overstated the association of the increase D-dimer levels of *V. vulnificus*-
infected patients to the occurrence of deep venous thrombosis.

We honestly hope that our responses will be enough to satisfy the reviewer.